# Optogenetic activation of Plexin-B1 reveals contact repulsion between osteoclasts and osteoblasts

Abhijit Deb Roy[1], Taofei Yin[1], Shilpa Choudhary[2], Vladimir Rodionov[1,3], Carol C. Pilbeam[2] & Yi I. Wu[1,4]

During bone remodelling, osteoclasts induce chemotaxis of osteoblasts and yet maintain spatial segregation. We show that osteoclasts express the repulsive guidance factor Semaphorin 4D and induce contact inhibition of locomotion (CIL) in osteoblasts through its receptor Plexin-B1. To examine causality and elucidate how localized Plexin-B1 stimulation may spatiotemporally coordinate its downstream targets in guiding cell migration, we develop an optogenetic tool for Plexin-B1 designated optoPlexin. Precise optoPlexin activation at the leading edge of migrating osteoblasts readily induces local retraction and, unexpectedly, distal protrusions to steer cells away. These morphological changes are accompanied by reorganization of Myosin II, PIP$_3$, adhesion and active Cdc42. We attribute the resultant repolarization to RhoA/ROCK-mediated redistribution of β-Pix, which activates Cdc42 and promotes protrusion. Thus, our data demonstrate a causal role of Plexin-B1 for CIL in osteoblasts and reveals a previously unknown effect of Semaphorin signalling on spatial distribution of an activator of cell migration.

[1] Richard D. Berlin Center for Cell Analysis and Modeling, University of Connecticut School of Medicine, 400 Farmington Avenue, Farmington, Connecticut 06030, USA. [2] New England Musculoskeletal Institute, University of Connecticut School of Medicine, Farmington, Connecticut 06030, USA. [3] Department of Cell Biology, University of Connecticut School of Medicine, Farmington, Connecticut 06030, USA. [4] Department of Genetics and Genome Sciences, University of Connecticut School of Medicine, Farmington, Connecticut 06030, USA. Correspondence and requests for materials should be addressed to Y.I.W. (email: yiwu@uchc.edu).

In multicellular organisms, migrating cells respond to attractive or repulsive cues to precisely control speed and directionality, and reach their destination with spatial and temporal accuracy. Contact inhibition of locomotion (CIL) describes the repulsive effect on a migrating cell upon contact with another cell[1]. CIL has been implicated in many physiological and pathological phenomena such as embryonic development, tissue patterning, collective migration and cancer metastasis. Although CIL has long been observed in vitro[2–4] and recently in vivo[5–9], the precise spatial and temporal dynamics of the underlying signalling remain obscure.

Guidance molecules such as semaphorins have been proposed to mediate repulsion in vivo[10–13]. In vertebrates, there are five classes of semaphorins, numbered from 3 to 7, each comprising of several members[14]. Excluding Semaphorin 3s, which are secreted, most semaphorins are either transmembrane or membrane-tethered proteins, suggesting a prerequisite role of cell–cell contact in their signalling. Semaphorins signal primarily through the Plexin family of single-pass transmembrane receptors[14]. Semaphorins interact with plexins through their respective extracellular sema domains[15,16]. The binding is thought to relieve autoinhbition of the receptor and mediate clustering of plexin intracellular domains[17–19]. The intracellular region of plexins contains a RasGAP domain that inactivates R-Ras[20]. B family plexins contain a PDZ-binding domain (PBD) at their carboxy terminus through which they associate with two PDZ-domain containing RhoGEFs, PDZ-RhoGEF (PRG) and Leukemia-associated RhoGEF (LARG), and activate RhoA[21–23].

During bone remodelling osteoclasts release different factors to mediate chemotaxis of osteoblasts to the site of resorption[24–27]. Semaphorin 4D (Sema4D), expressed by osteoclasts, has been shown recently to regulate osteoblast-mediated bone formation[28,29]. Ablation of Sema4D or Plexin-B1 reduced the spacing between osteoclasts and osteoblasts in vivo[28], suggesting that osteoclasts may also repel osteoblasts, which is contrary to the paradigm of chemoattraction between the cells. Whether osteoclasts induce repulsion in osteoblasts has not been widely recognized and the molecular pathways that may mediate such response have not been explored.

In the current study, we demonstrate that osteoclasts repel osteoblasts upon contact and determine that CIL between these cells is dependent on Sema4D–Plexin-B1 signalling. To elucidate the spatiotemporal dynamics of Plexin-B1 downstream effectors that mediate repulsion, we develop an optogenetic tool (optoPlexin) to initiate Plexin-B1 signalling at precise times and subcellular locations. In contrast to a collapse phenotype upon whole-cell activation, localized optoPlexin stimulation induces a coordinated retraction at the site of illumination and protrusions at distal regions. Similar to CIL with osteoclasts, the repulsion phenotype induced by precise optoPlexin stimulation does not alter the inherent motility of the cells. Employing biosensors for Myosin II, phosphatidylinositol(3,4,5)P$_3$ (PIP$_3$) and Rho GTPases including RhoA, Cdc42 and Rac1, we define the spatial and temporal regulation of signalling downstream of Plexin-B1. Finally, we identify a novel mechanism by which Plexin-B1 coordinates cell repolarization through RhoA-ROCK-mediated redistribution of β-Pix.

## Results

### Osteoclasts induce CIL in osteoblasts. To examine whether osteoclasts affect osteoblast migration, we isolated primary bone marrow macrophages (BMMs) and primary calvarial osteoblasts (POBs) from mice. Upon differentiation in culture, the BMMs formed multinucleated osteoclasts and were then overlaid with POBs. Similar to earlier reports, we observed that

the osteoclasts pushed the obsteoblasts away upon contact[4,30]. There were physical gaps between these two cell types and the osteoblasts lacked lamellipodial protrusions towards the osteo-clasts (Supplementary Fig. 1a and Supplementary Movie 1). To investigate how osteoblast migration was altered, we performed a 'wound-healing' assay (Fig. 1a,b) in which POBs or MC3T3-E1 cells were seeded next to osteoclasts. On removing the insert, both POBs (Supplementary Movie 2) and MC3T3-E1 cells (Supplementary Movie 3) formed lamellipodial protrusions and migrated towards the osteoclasts. Upon contact with osteoclasts, these protrusions rapidly collapsed, followed by formation of new protrusions away from the site of contact and a change in migration direction (Fig. 1d, Supplementary Fig. 1b and Supplementary Movies 2 and 3). In MC3T3-E1 cells, the protrusions collapsed on average 6 min after initiation of contact with osteoclasts and new distal protrusions formed about a minute after (Supplementary Fig. 2a). Although the migration speed before and after contact remained unaltered (Supplementary Fig. 2b), the contact acceleration index (Cx, see Methods for details) showed a clear reversal of migration direction with respect to the trajectory before contact (negative values). The Cx approached zero before contact, indicating directional persistence (Fig. 1h,i). Thus, our data indicate that osteoclasts induce CIL in obsteoblasts without compromising their intrinsic motility.

### Plexin-B1 mediates osteoclast-induced CIL in osteoblasts. Sema4D is expressed by BMMs upon differentiation to osteoclastic lineage using receptor activator of nuclear factor κ-B ligand (RANKL). Consistent with previous reports[28,29], Sema4D expression was undetectable in BMMs but was markedly increased during RANKL-mediated osteoclastogenesis (Supplementary Fig. 3a). In support of a role of Sema4D in CIL, BMMs failed to induce collapse of protrusions or change in direction of migration in MC3T3-E1 cells (Supplementary Fig. 4a,b and Supplementary Movie 4). Both POBs and MC3T3-E1 cells expressed Plexin-B1 and no significant changes were observed in messenger RNA or protein levels upon differentiation (Fig. 1c and Supplementary Fig. 3b,c). To examine whether CIL between MC3T3-E1 and osteoclasts was mediated by Plexin-B1, we conducted knockdown experiments with several independent short hairpin RNA (shRNA) lentivirus, small interfering RNA (siRNA) pools, as well as CRISPR/Cas9-mediated knockout experiments. Despite our best efforts, only modest silencing of protein expression (Supplementary Fig. 3d,e) was achieved with siRNA and shRNA without compromising cell motility. In contrast, the CRISPR/Cas9-mediated knockout approach (see Methods for details) completely eliminated Plexin-B1 expression (Fig. 1c). Two Plexin-B1-null MC3T3-E1 clones, KO1 and KO2, were used in cell migration assays. Cas9 expression alone did not affect Plexin-B1 expression (Fig. 1c), nor did it alter osteoclast-mediated contact repulsion of MC3T3-E1 cells (Fig. 1f, Supplementary Fig. 4c and Supplementary Movie 5). KO1 and KO2 cells failed to show collapse of protrusions upon contact with osteoclasts and maintained contact with obsteoclasts for prolonged periods of time (Fig. 1e,f, Supplementary Fig. 4d and Supplementary Movies 6 and 7). Furthermore, they did not migrate away from the osteoclasts (Fig. 1g). Finally, comparison of Cx values (Fig. 1i) indicated that in contrast to wild-type and Cas9 cells, KO1 and KO2 cells failed to undergo repulsion after contact. Thus, our data demonstrated that Plexin-B1 is required for CIL in osteoblastic MC3T3-E1 cells.

### Development of an optogenetic tool for the Plexin receptor. To activate Plexin-B1 in osteoblasts, we initially attempted several Sema4D ligand-based approaches. Perfusion of soluble

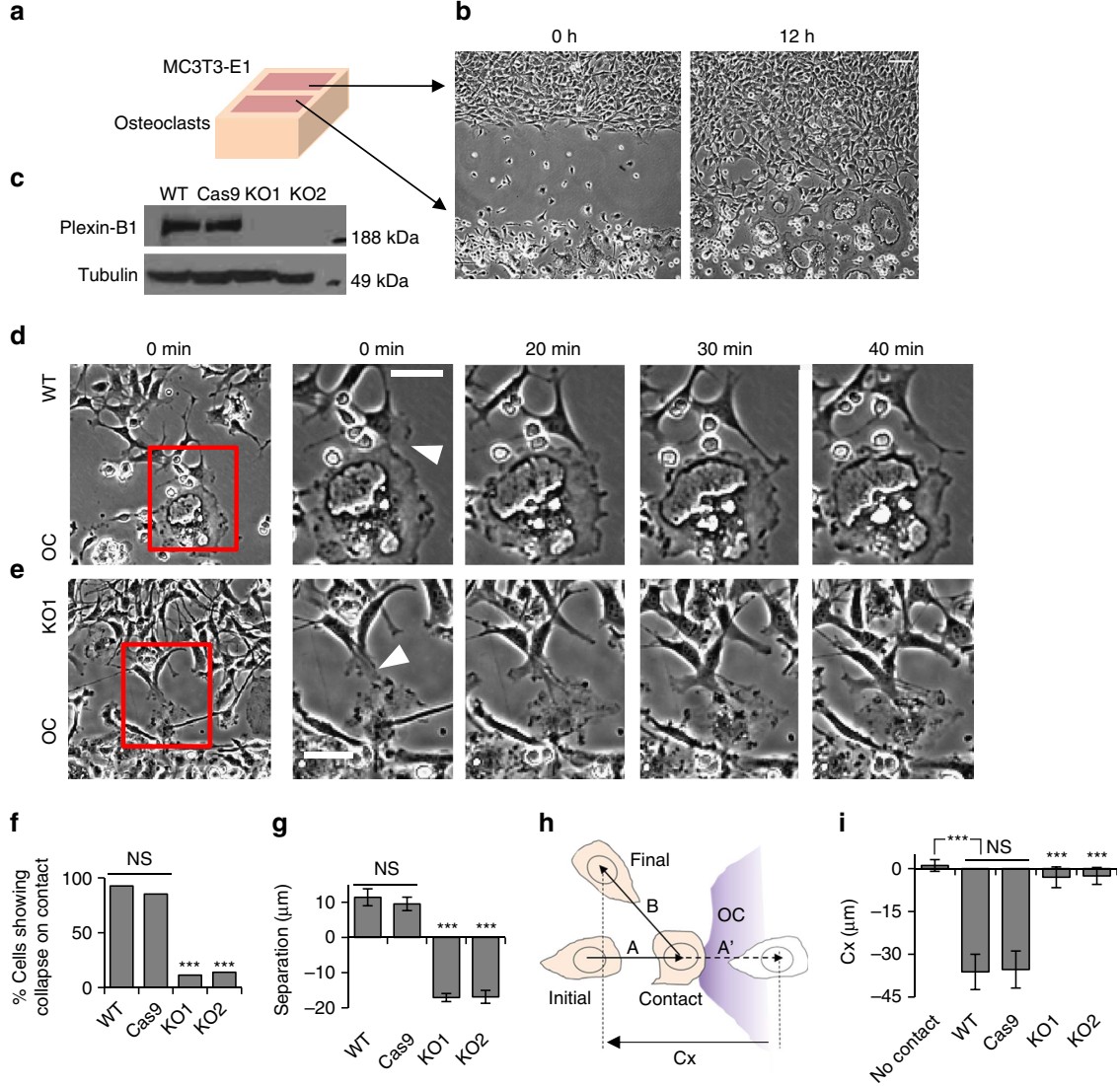

**Figure 1 | Plexin-B1 mediates CIL between osteoclasts and osteoblastic cells. (a)** Cartoon representation of culture inserts used for co-culture of osteoclasts with primary osteoblasts or MC3T3-E1 cells. **(b)** Phase-contrast images acquired immediately and 12 h after lifting the culture insert showing migration of MC3T3-E1 cells towards osteoclasts. Scale bar, 50 μm. **(c)** Comparison of Plexin-B1 expression in wild-type (WT), Cas9, KO1 and KO2 MC3T3-E1 cells. Tubulin was used as a loading control. **(d,e)** Phase-contrast images showing a MC3T3-E1 WT **(d)** or MC3T3-E1 KO1 **(e)** cell in contact with an osteoclast (OC). Insets are magnified to show changes in cell morphology and migration between time of contact and 20, 30 and 40 min after contact. Scale bar, 50 μm. White triangles point to the site of cell–cell contact. **(f)** Percentage of WT, Cas9, KO1 and KO2 MC3T3-E1 cells that show collapse of protrusions within 30 min after contact with an osteoclast. $n = 209$–301. **(g)** Separation between MC3T3-E1 cells and osteoclasts 40 min after contact. $n = 41$–45. **(h)** Cartoon describing Cx. **(i)** Cx values for WT cells, Cas9, KO1 and KO2 MC3T3-E1 cells without (WT only) and after contact with osteoclasts. $n = 20$. For **g,i**, means ± s.e.m. are shown. $***P < 0.001$ and $*P < 0.05$, NS, not significant, Student's $t$-test.

Sema4D-Fc ligand induced RhoA activation in MC3T3-E1 cells and cell collapse as indicated by loss of protrusions and decrease in total cell area (Supplementary Fig. 5). Contact between osteoclasts and osteoblasts may initiate Sema4D-Plexin-B1 signalling specifically at the site of contact. To understand the role of Plexin-B1 in CIL, we locally pumped Sema4D-Fc near protrusions of POBs using a microinjection pipette (Supplementary Fig. 6). The cells responded with a blunted protrusion or retraction locally and four out of five times produced a new protrusion(s) in distal regions, which is consistent with CIL. In contrast, no significant morphological changes were observed with control IgG1 (Supplementary Fig. 6). As it is difficult to confine soluble ligand in the medium, we used immobilized Sema4D-Fc on silica beads as localized sources of Sema4D

stimulation. Despite rapidly clustering Plexin-B1 on the plasma membrane (Supplementary Fig. 7 and Supplementary Movie 8), the Sema4D-Fc beads failed to induce RhoA activation, morphological changes or even recruit RhoGEFs (Supplementary Figs 8 and 9, see Methods for details).

To mimic local initiation of Sema4D signalling, we used an optogenetic approach to activate the plexin receptor. We took advantage of the observations that both the RasGAP activity of Plexin-B1 and its activation of RhoA require its localization to the plasma membrane, and that semaphorin-dependent clustering of plexin promotes its activation[17–19]. A recently characterized optogenetic module Cryptochrome-2 (Cry2) was used with a farnesylated Cryptochrome interacting basic helix–loop–helix 1 (CIB1-CAAX), to induce a plasma membrane translocation of

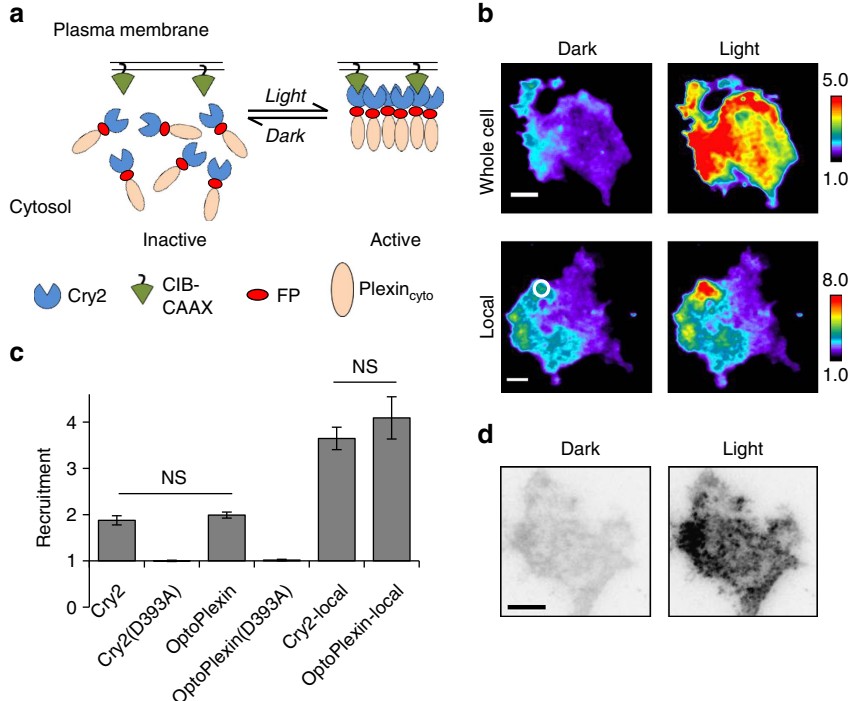

**Figure 2 | Design and characterization of optoPlexin.** (**a**) Cartoon representation of the optoPlexin design (see main text). FP, fluorescent protein; Plexin$_{cyto}$, cytosolic domain of Plexin-B1. (**b**) TIRF images showing the plasma membrane recruitment of mCherry-optoPlexin in COS-7 cells after 1 min of whole cell or local illumination (440 nm and 50 ms pulse at 0.1 Hz, unless otherwise stated). Scale bar, 10 μm. White circle, region of illumination. (**c**) Quantification of the recruitment levels of Cry2, Cry2(D393A), optoPlexin and optoPlexin(D393A) on whole cell or localized blue light illumination in COS-7 cells. $n = 12$–20, means ± s.e.m. (**d**) TIRF images showing the clustering of mCherry-optoPlexin after one minute of whole-cell illumination in COS-7 cells. Scale bar, 10 μm. NS, not significant, Student's $t$-test.

cytosolic proteins in response to blue light[31]. Furthermore, Cry2 oligomerizes in light and could be used to induce protein clustering[32]. We constructed a chimera protein containing Cry2 at the amino terminus, followed by a fluorescent protein (mCherry or mVenus), and finally the intracellular domain Plexin-B1 at the C terminus (Supplementary Fig. 10), which preserved its interaction with PRG and LARG. Full-length Cry2 was used to minimize background activity in the dark[31]. As the engineered Plexin-B1 was designed to be activated by light instead of Sema4D ligand, we named it optoPlexin (Fig. 2a).

To examine whether optoPlexin undergoes a plasma membrane translocation in response to light stimulation, we co-expressed mCherry-optoPlexin and CIB1-CAAX in COS-7 cells, and used total internal reflection fluorescence (TIRF) microscopy to gauge the membrane association of optoPlexin. Upon blue light (440 nm light emitting diode, LED) illumination over the entire cell, mCherry intensity increased immediately after the first pulse of illumination. The rate of increase appeared to be at a similar time scale as Cry2 alone, and with our illumination protocol of 10 s intervals, reached half maximal intensity within three pulses, indicating a rapid recruitment of optoPlexin to the plasma membrane (Fig. 2b top panel, Supplementary Fig. 11 and Supplementary Movie 9 left panel). The average increase in the intensity of optoPlexin and optoPlexin mutants (Supplementary Fig. 10) were close to twofold, indistinguishable from that of Cry2 alone (Fig. 2c and Supplementary Fig. 12). In addition, optoPlexin formed clearly visible fluorescent aggregates, indicating clustering of optoPlexin (Fig. 2d and Supplementary Fig. 13). A flavin adenine dinucleotide (FAD)-deficient mutant of Cry2 (D387A) that does not absorb blue light[33] failed to recruit optoPlexin (Supplementary Movie 9 right panel). As a lack of FAD may potentially

compromise the folding of Cry2 or optoPlexin, we also examined an alternate mutation of Cry2 (D393A), which preserves the FAD-binding pocket but eliminates the proton donor (Asp[393]) for FAD and blocks signal transduction of crytochrome in plants[34,35]. No membrane recruitment took place for either Cry2 alone or optoPlexin carrying the D393A mutation (Fig. 2c), demonstrating that the observed effects were specific to the photoreaction of Cry2. More effective recruitment, an average induction of fourfold within the region of illumination (Fig. 2b,c), was observed when illumination was limited to a 5 μm-diameter circle within the cells. These data demonstrated that optoPlexin can be efficiently recruited to the plasma membrane and cluster in response to light stimulation.

**Robust activation of RhoA by optoPlexin on light stimulation.** Upon Sema4D stimulation, Plexin-B1 interacts with and activates PRG or LARG through its C-terminal PBD and consequently activates RhoA[12,21,23]. To test whether optoPlexin could interact with PRG and LARG, we examined the membrane recruitment of these two RhoGEFs by co-expressing mVenus-PRG or mVenus-LARG in COS-7 cells. Upon whole-cell illumination, we observed a rapid, concurrent co-recruitment of PRG (Fig. 3a,b and Supplementary Movie 10) or LARG (Fig. 3b) with optoPlexin to the plasma membrane. The kinetics of PRG recruitment and dissociation from the membrane closely followed that of optoPlexin. Co-recruitment of PRG was reversible upon pausing blue light illumination and repeatable with a second round of illumination (Supplementary Fig. 11). An increase in fluorescence of 26% ± 3% and 60% ± 6% (mean ± s.e.m.) relative to that of optoPlexin were observed for PRG and LARG,

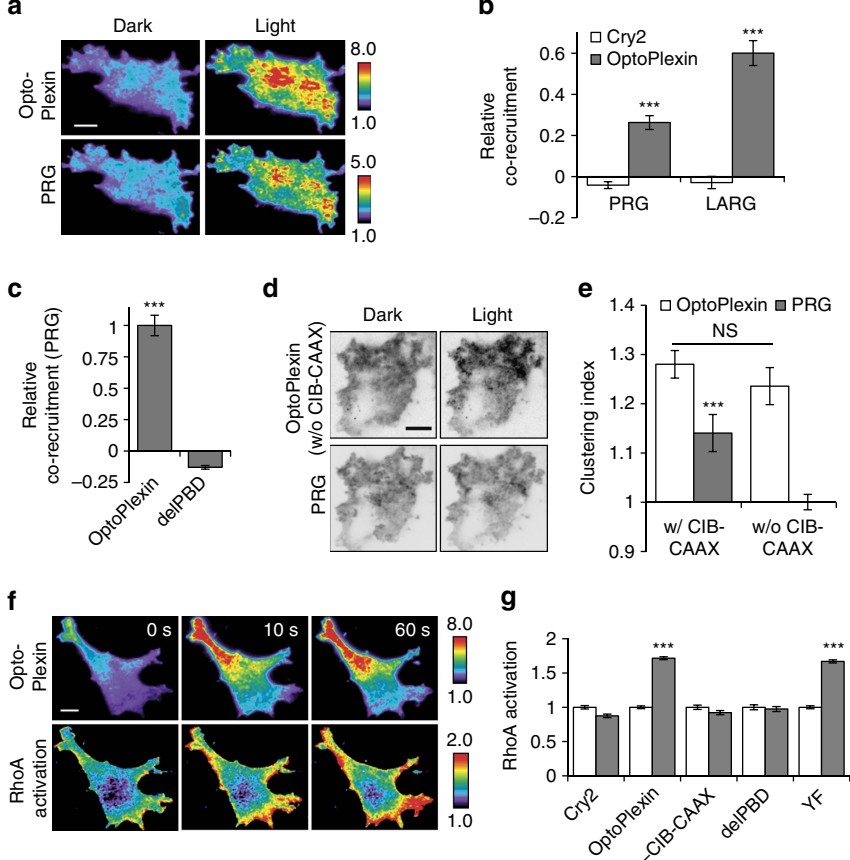

**Figure 3 | OptoPlexin activates RhoA through membrane recruitment of RhoGEFs.** (**a**) TIRF images showing the membrane recruitment of mVenus-PRG along with mCherry-optoplexin after one minute of whole cell illumination in COS-7 cells. Scale bar, 10 μm. (**b**) Relative co-recruitment of mVenus-PRG or LARG by Cry2 and optoplexin upon whole-cell illumination in COS-7 cells, $n = 13$–19. (**c**) Normalized relative co-recruitment of mVenus-PRG by optoplexin ($n = 19$) and optoplexin-delPBD ($n = 13$). (**d**) TIRF images showing clustering of optoplexin and lack of that with mVenus-PRG, in the absence of CIB-CAAX after 1 min of whole-cell illumination in COS-7 cells. Scale bar, 10 μm. (**e**) Quantification of clustering of optoplexin and PRG in the presence ($n = 19$) or absence of CIB-CAAX ($n = 13$). (**f**) Induction of mCherry-optoplexin upon ratiometric imaging of DORA-RhoA biosensor using TIRFM. Scale bar, 10 μm. (**g**) RhoA activation in COS-7 cells expressing Cry2, optoplexin, optoplexin-delPBD or optoplexin-YF upon whole-cell illumination, $n = 18$–32 cells. For **b,c,e,g**, means ± s.e.m. are shown.***$P < 0.001$ and *$P < 0.05$, NS, not significant, Student's $t$-test.

respectively (Fig. 3b), suggesting an efficient recruitment of RhoGEFs to the plasma membrane by optoplexin. As optoplexin may also interact with endogenous PRG and LARG, the extent to which signals were transmitted from optoplexin to these two RhoGEFs were most probably underestimated. When Cry2 alone was used as a control, no membrane recruitment of these two RhoGEFs was detected (Fig. 3b). Illumination of cells expressing a PBD-truncated mutant (delPBD) (Supplementary Fig. 10) showed no increase of PRG levels on the plasma membrane (Fig. 3c), indicating that PBD–PDZ interaction mediates RhoGEF association with optoplexin. The interaction between optoplexin and PRG did not appear to be constitutive. Upon illumination, PRG formed quantifiable aggregates (see Methods for details) along with optoplexin in addition to membrane recruitment (Supplementary Fig. 13). When we omitted the CIB-CAAX from optoplexin, the clustering of optoplexin remained visible using TIRF imaging but we did not detect any clustering of PRG (Fig. 3d,e), suggesting that membrane translocation is critical for activation of optoplexin.

To directly test whether RhoA was activated upon optoplexin stimulation, we employed the Dora-RhoA biosensor described previously[36]. Owing to overlapping wavelengths, the excitation light for Dora-RhoA fluorescence resonance energy transfer (FRET) sensor was sufficient to induce rapid membrane recruitment of

mCherry-optoplexin in COS-7 cells (Fig. 3f upper panel and Supplementary Fig. 14). Although this prevented us from accurately capturing the level of RhoA activation before illumination, a consistent increase of RhoA activation was readily detectable (Fig. 3f lower panel and Supplementary Fig. 14), with an average of 70% induction of RhoA activation within the first three acquisitions (Fig. 3g and Supplementary Fig. 14b). Spatial pattern of RhoA activation did not accurately match with optoplexin recruitment levels, potentially due to spatial regulation of RhoA activation through other factors. Consistent with being further downstream of Plexin-B1, the activation of Dora-RhoA exhibited a delay (~20 s) in comparison with the membrane recruitment of optoplexin (Supplementary Fig. 14b). Importantly, the effects were specific, because no RhoA activation was detected when Cry2 alone or optoplexin-delPBD was used, or when CIB-CAAX was omitted (Fig. 3g). Sema4D-dependent ErbB2 association with Plexin-B1 mediates downstream RhoA activation in the osteoblasts[28,37,38]. However, neither inhibiting ErbB2 with Erlotinib[39] nor mutations designed to abrogate ErbB2 regulation of Plexin-B1 (ref. 38) (optoplexin-YF) (Supplementary Fig. 10) had any negative impact on PRG co-recruitment (Supplementary Fig. 15a,b); in fact, we observed increased PRG recruitment in both cases (Supplementary Fig. 15c). OptoPlexin-YF induced RhoA activation at levels comparable to optoplexin wild type (Fig. 3g). Based on these

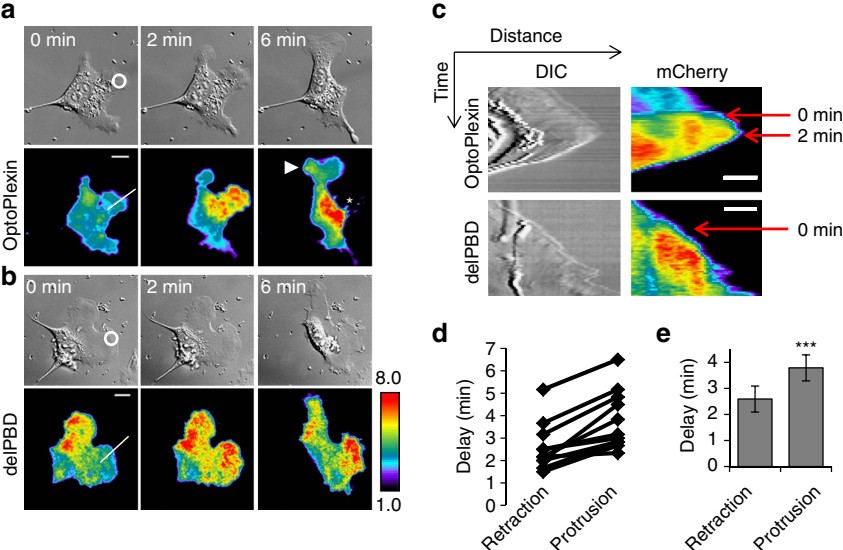

**Figure 4 | OptoPlexin induces CIL upon local activation.** MC3T3-E1 cells expressing mCherry-optoPlexin (**a**) or mCherry-optoPlexin-delPBD (**b**) were locally illuminated as indicated (white circle). The DIC and TIRF images were shown to illustrate the morphological changes and local membrane recruitment. Scale bar, 10 μm. Asterisk, retraction. Arrowhead, induced protrusion. (**c**) Kymographs showing cell border progression upon local activation of optoPlexin or optoPlexin-delPBD. Reference lines for the kymographs are shown in white in **a**,**b**. Scale bar, 10 μm. (**d**) Pairwise comparison of the delay in initiation of retractions and protrusions upon local optoPlexin activation. (**e**) Mean delay in initiation of local retractions and distal protrusions on local optoPlexin activation. $n = 11$ cells, means ± s.e.m. are shown. ***$P < 0.001$ and *$P < 0.05$. NS, not significant, Student's $t$-test.

results, we concluded that optoPlexin exhibits minimal background activity in the dark and activates RhoA robustly upon illumination.

**Induction of CIL upon local activation of optoPlexin.** To precisely control where Plexin-B1 is activated, we expressed optoPlexin and CIB-CAAX in MC3T3-E1 cells, and illuminated a 5 μm circular region in protrusions to mimic contact with Sema4D-expressing osteoclasts. As optoPlexin continued to accumulate, the illuminated protrusion collapsed and started to retract (Fig. 4a, indicated with an asterisk, and Supplementary Movie 11). The effects of optoPlexin were better illustrated in kymograph analyses where the accumulation of optoPlexin and the sharp transition of the cell border became apparent (Fig. 4c). Despite similar enrichment in the illuminated protrusions, optoPlexin-delPBD failed to induce retraction (Fig. 4b,c and Supplementary Movie 13), suggesting RhoA activation is required for optoPlexin-induced retraction. Among the optoPlexin-expressing cells, the average delay from the start of illumination to initiation of retraction was 2.5 ± 0.5 min (mean ± s.e.m., Fig. 4d). Interestingly, in addition to retraction, we observed either formation of a new protrusion(s) (Fig. 4a, indicated with an arrowhead) or enhancement of an existing protrusion(s) distal to the site of illumination. They appeared immediately after the initiation of retractions, with an average time delay of 75 s (Fig. 4d,e), suggesting that they may be mechanistically coupled. Even though optoPlexin induced retraction locally (Fig. 5a), the overall cell area remain unchanged (Fig. 5b).

To quantify the effects of optoPlexin on motility, we superimposed images of the cell boundary before and 7.5 min after illumination and determined the relative orientations of the illumination site, retraction and protrusion with respect to the centroid of the cell immediately before illumination (Fig. 5d). Retractions were closely aligned with the orientation of the illumination site (Fig. 5e,g), whereas new protrusions were randomly distributed (Fig. 5f,g). As the centroid is dictated by the original cell shape, thereby influencing the relative orientations of protrusion and retraction, we also measured the linear

distance between protrusive or retracting regions and the site of illumination (Fig. 5d). Retractions took place within 10 μm of the site of illumination, as compared with protrusions, which were significantly farther from the site of illumination (Fig. 5h). Persistent illumination in protrusions effectively repolarized the cells and and significantly altered their direction so that the cells migrated away from the region of illumination (Supplementary Fig. 16a and Supplementary Movie 12). In some cases we had to change the region of illumination to new protrusions to effectively guide the cells. Tracing the displacements of the centroid and the nucleus of one such cell showed a gradual change in centroid direction preceding a sharp change in the direction of nuclear migration (Supplementary Fig. 16b,c). Consistent with our observations with osteoclast co-culture (Supplementary Fig. 2b), we did not observe significant changes in cell centroid or nuclear velocities in MC3T3-E1 cells upon optoPlexin stimulation (Fig. 5c and Supplementary Fig. 17). By contrast, whole-cell illumination of optoPlexin-expressing cells led to collapse of all existing protrusions and hindered motility (Supplementary Movie 14), similar to the response on bath application Sema4D-Fc (Supplementary Fig. 5a,b). Thus, localized Plexin-B1 signalling through optoPlexin stimulation revealed a CIL-like phenotype in MC3T3-E1 cells that had not been demonstrated previously.

**Both RhoA and RasGAP pathways are required for CIL.** To determine which pathway(s) downstream of optoPlexin mediates CIL, we quantified the changes in protusive areas induced by optoPlexin activation. We already showed that the delPBD mutant failed to inhibit protrusion (Figs 4b and 5a, and Supplementary Movie 13). A specific ROCK inhibitor Y-27632 also abrogated optoPlexin-mediated retractions (Fig. 5a), further validating the role of RhoA-ROCK signalling in Plexin-B1-mediated retractions. In addition, we tracked the signalling molecules upstream or downstream of RhoA. We locally activated mCherry-optoPlexin in cells coexpressing mVenus-PRG and found PRG was recruited immediately to the spot where

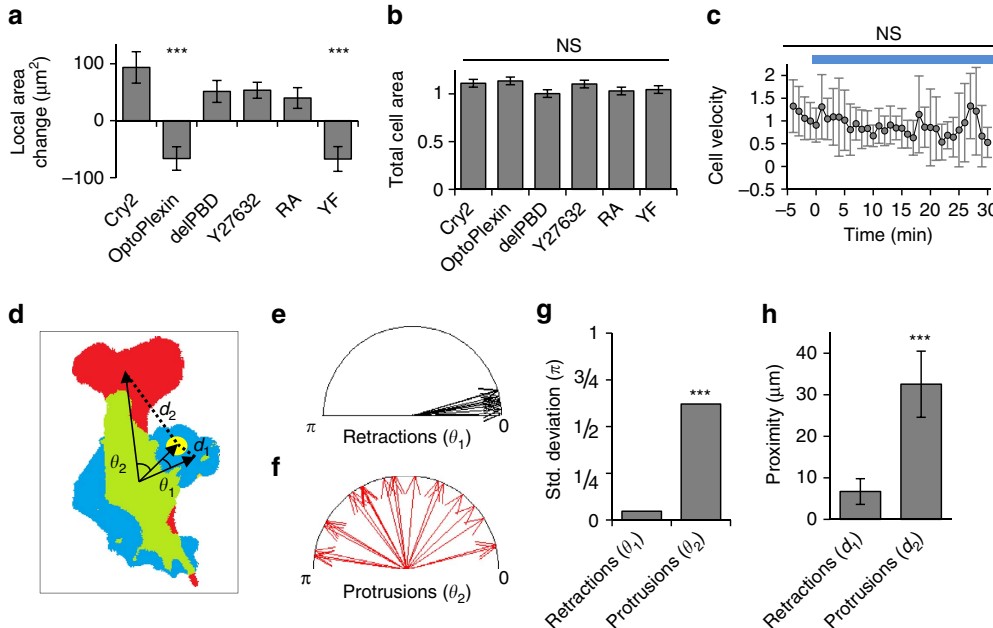

**Figure 5 | Effects of local activation of optoPlexin on cell morphology and motility.** Effects on the illuminated protrusions (**a**), measured in a 50 μm-diameter circle centred at the region of illumination, and total cell areas (**b**) after 7.5 min of local illumination in MC3T3-E1 cells expressing Cry2, optoPlexin, optoPlexin-delPBD, optoPlexin and pretreated with 10 μM Y-27632 ROCK inhibitor, and optoPlexin-RA, $n = 9$–14 cells. (**c**) Cell centroid velocities at different times upon local activation of optoPlexin in MC3T3-E1 cells. Blue line, illumination at 440 nm, $n = 21$ cells. (**d**) The locally induced retraction (blue) and the distal protrusion (red) after pulses of illumination (yellow) of an optoPlexin-expressing cell was illustrated in a morphology diagram. The overlapping cell area before and after illumination was labeled in green. Based on the centroids of these colour-coded regions, $\theta_1$ and $\theta_2$ were used to describe the angles of retraction and protrusion relative to the direction from the centroid of the cell at time 0 to the centre of illumination, respectively. Similarly, $d_1$ and $d_2$ indicated their centroid distances to the region of illumination, respectively. (**e,f**) Angles of retraction and protrusion induced by local activation of optoPlexin in MC3T3-E1 cells and (**g**) s.d. of their distribution, $n = 14$ cells. (**h**) Proximity of retractions and protrusions induced by local activation of optoPlexin in MC3T3-E1 cells to the region of illumination, $n = 14$ cells. For **a,b,c,h**, means ± s.e.m are shown, ***$P < 0.001$ and *$P < 0.05$, NS, not significant, Student's $t$-test. For **g**, ***$P < 0.001$ and *$P < 0.05$, NS, not significant, F-test.

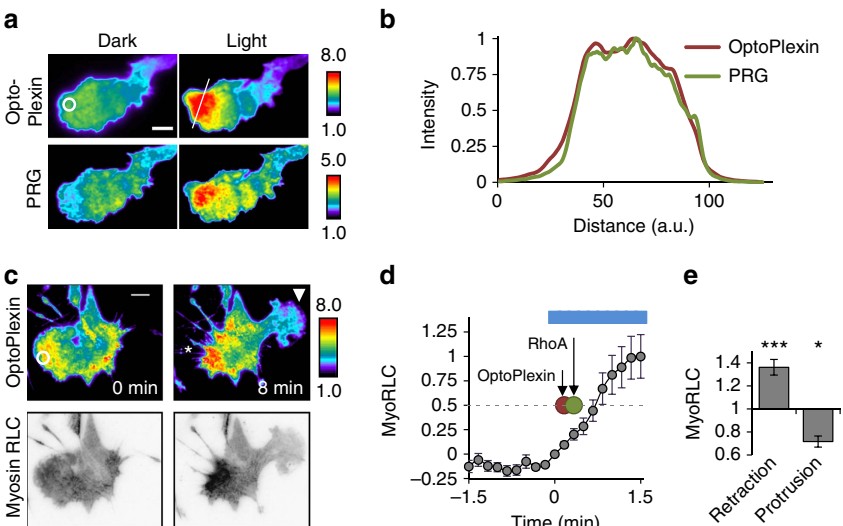

**Figure 6 | OptoPlexin induces CIL through RhoA-mediated pathways.** (**a**) TIRF images showing localized membrane recruitment of mVenus-PRG with mCherry-optoPlexin in MC3T3-E1 cells upon local illumination. White circle, region of illumination. Scale bar, 10 μm. (**b**) Normalized intensities of mCherry-optoPlexin and mVenus-PRG after local illumination in a linescan analysis. Reference line for the linescan is shown in the right top panel of **a**. (**c**) TIRF images showing a polarized distribution of mCherry-MyoRLC upon local activation of mVenus-optoPlexin in MC3T3-E1 cells. White circle, region of illumination. Asterisk, induced retraction. Arrowhead, induced protrusion. Scale bar, 10 μm. (**d**) Temporal changes in average mCherry-MyoRLC intensity in a 50 μm diameter circle around the region of illumination (white circle) in MC3T3-E1 cells upon localized activation of mVenus-optoPlexin. Red and green circles indicate $t_{1/2}$ for optoPlexin and RhoA inductions, respectively, as estimated from Supplementary Fig. S4b. Blue line, illumination at 440 nm. (**e**) Relative intensities of mCherry-MyoRLC in retractions and new protrusions induced by local activation of mVenus-optoPlexin, normalized to the whole cell average. For **d,e** $n = 12$ cells, means ± s.e.m. ***$P < 0.001$ and *$P < 0.05$, NS, not significant, Student's $t$-test.

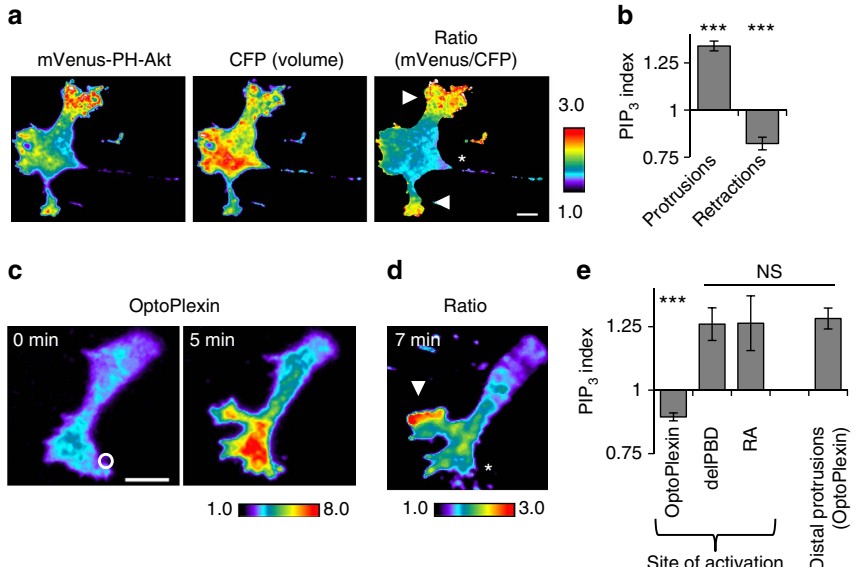

**Figure 7 | Local activation of optoPlexin repolarizes PIP₃.** (**a**) TIRF images of mVenus-PH-Akt and CFP (volume marker) in an MC3T3-E1 cell were processed to generate a ratiometric image (mVenus/CFP) that displays the membrane enrichment of mVenus-PH-Akt, indicative of $PIP_3$ concentration. Asterisk, retraction. Arrowheads, protrusions. Scale bar, 10 μm. (**b**) $PIP_3$ distribution in protrusions and retractions in MC3T3-E1 cells, as measured by membrane enrichment mVenus-PH-Akt normalized to the whole cell average, $n = 9$ cells (**c**) TIRF images showing local induction of mCherry-optoPlexin in a MC3T3-E1 cell expressing mCherry-optoPlexin, mVenus-PH-Akt and CFP. Upon induction of CIL, mVenus and CFP images were obtained and processed to generate ratiometric image (**d**) indicating $PIP_3$ distribution. White circle, region of illumination. Scale bar, 10 μm. Asterisk, induced retraction. Arrowhead, induced protrusion. (**e**) Quantification of $PIP_3$ indices in a 50 μm diameter circle centred at the region of illumination upon local activation of optoPlexin, optoPlexin-delPBD and optoPlexin-RA; and in distal protrusions induced by local activation of optoPlexin, normalized to whole cell average, $n = 8$–12 cells. For **b**,**e** means ± s.e.m. are shown. ***$P < 0.001$ and *$P < 0.05$, NS, not significant, Student's $t$-test.

optoPlexin accumulated (Fig. 6a and Supplementary Movie 15). The line-scan profile of PRG resembled that of optoPlexin very closely (Fig. 6b), which is consistent with a physical interaction between these two proteins. Myosin II regulatory light chain (MyoRLC) is phosphorylated by ROCK downstream of RhoA, which leads to its association with actin filaments. We used mCherry-MyoRLC in TIRF imaging to track the immobilized myoRLC as an indicator of myosin II activation[40]. Upon local activation of mVenus-optoPlexin, we observed accumulation of mCherry-myoRLC at the site of illumination, but with an apparent delay (Fig. 6c,d and Supplementary Movie 16). Interestingly, the distal protrusions induced were associated with substantially decreased myoRLC, suggesting a depletion of myosin activity in these nascent protrusions (Fig. 6c,e). An optoPlexin mutant (optoPlexin-RA) (Supplementary Fig. 10) lacking RasGAP activity also failed to induce retraction (Fig. 5a, Supplementary Fig. 18 and Supplementary Movie 17). The response of MC3T3-E1 cells upon local activation of optoPlexin-YF mutant was indistinguishable from that of optoPlexin (Fig. 5a, Supplementary Fig. 19 and Supplementary Movie 18). In all cases where retraction was not induced, protrusions also failed to form in distal regions. Thus, both RhoA and RasGAP pathways are required for optoPlexin to induce CIL.

**Local activation of optoPlexin repolarizes PIP₃.** Phosphoinositides, in particular $PIP_3$, are extensively involved in regulation of cell polarity and motility[41,42]. To examine whether $PIP_3$ is regulated in migration of MC3T3-E1 cells, we employed the PH domain of Akt (PH-Akt) to monitor the spatial distribution of $PIP_3$ (ref. 44). Ratiometric imaging of mVenus-PH-Akt and a cytosolic mCerulean volume maker were imaged using TIRF microscopy and their ratio (mVenus/mCerulean) was used to

indicate the levels of $PIP_3$ enrichment. Consistent with a role of $PIP_3$ in migration, we observed accumulation of $PIP_3$ at the leading edge and low levels in retracting areas of migrating MC3T3-E1 cells (Fig. 7a,b). Because of the overlapping wavelengths between optoPlexin activation and mCerulean excitation, to investigate spatial regulation of $PIP_3$ during optoPlexin-mediated CIL, we activated mCherry-optoPlexin at protrusions first and acquired a pair of mVenus and mCerulean images upon induction of distal protrusions. In contrast to an elevated $PIP_3$ seen in unperturbed protrusions, a substantial reduction of $PIP_3$ was observed in the area of optogenetic illumination. Furthermore, we observed an acute accumulation of PH-Akt in the newly formed distal protrusions (Fig. 7c,d). Neither optoPlexin-delPBD nor optoPlexin-RA induced significant decrease of local $PIP_3$ (Fig. 7d) upon at least 7.5 min of local activation. Thus, our data supported the requirement of both RhoA activation and RasGAP activity of Plexin-B1 in regulating $PIP_3$ in osteoblasts.

**OptoPlexin spatially coordinates Cdc42 activity in CIL.** Small GTPases Rac1 and Cdc42 are potent inducers of actin polymerization and migration. Employing Dora-Cdc42 and Dora-Rac1 biosensors[36], we observed elevated Cdc42 activation at protrusions in migrating MC3T3-E1 cells, whereas Rac1 activity was less polarized (Supplementary Fig. 20). To examine how optoPlexin may affect Cdc42 and Rac1 activities, we co-expressed mCherry-optoPlexin with a Dora-Cdc42 or Dora-Rac1 sensor in MC3T3-E1 cells. We activated optoPlexin exclusively in a protrusion where Cdc42 was expected to be active. To avoid activating optoPlexin globally, we acquired sensor images when distal protrusion(s) were induced (Fig. 8a,b). We did not detect any active Cdc42 in the illuminated and retracting area, suggesting local inhibition of Cdc42 activity (Fig. 8b,e). In contrast,

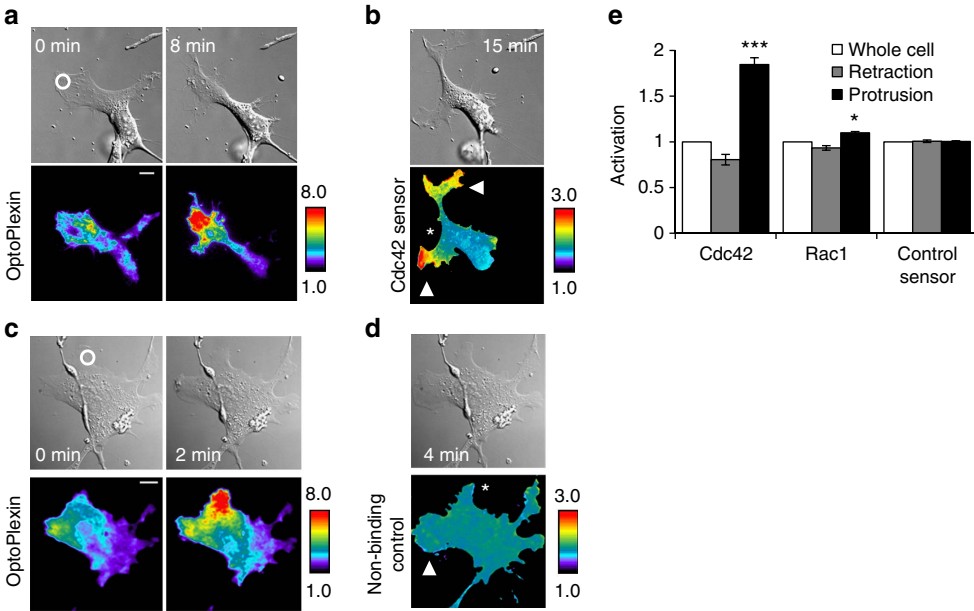

**Figure 8 | Local activation of optoPlexin spatially coordinates Cdc42 and Rac1 activities.** DIC and TIRF images showing local accumulation and activation of mCherry-OptoPlexin in MC3T3-E1 cells co-expressing a Dora-Cdc42 biosensor (**a**) or its non-binding control (**c**). Upon induction of CIL, a pair of FRET and CFP images were acquired using TIRFM and processed to generate a ratio image for the Dora-Cdc42 sensor (**b**) and its non-binding control (**d**). White circle, region of illumination. Scale bar, 10 μm. Asterisk, induced retraction. Arrowhead, induced protrusion. (**e**) Quantificaiton of the spatial distribution of active Cdc42 and Rac1 in MC3T3-E1 cells after induction of CIL, $n = 9$–12 cells, means ± s.e.m. are shown. ***$P < 0.001$ and *$P < 0.05$, NS not significant, Student's $t$-test.

the new protrusions in distal regions showed marked activation of Cdc42. Moreover, a non-binding control sensor exhibited low ratio values and minimal variation across the cell (Fig. 8c–e), suggesting that the Cdc42 polarization was not due to a volume artifact. Using the Dora-Rac1 sensor, we found that Rac1 was also activated in the distal protrusion(s), albeit to a substantially reduced level (Fig. 8e and Supplementary Fig. 21). Thus, our data indicated that spatial and temporal activation of Cdc42 and possibly Rac1 may coordinate the opposing effects between local and distal regions upon optoPlexin activation.

**OptoPlexin redistributes β-Pix to distal regions during CIL.** To understand how optoPlexin may regulate Cdc42 and Rac1, we focused on a ubiquitously expressed RhoGEF β-Pix, which activates Cdc42 and Rac1 (refs 43,44) and mediates cross-talk between RhoA and Cdc42/Rac1. In migrating MC3T3-E1 cells, β-Pix is predominantly associated with nascent adhesions and focal complexes in protrusions rather than with focal adhesions in stationary or retracting areas (Supplementary Fig. 22a–c). Consistent with previous reports[45], inhibition of RhoA-ROCK signalling using ROCK inhibitor Y-27632 promoted association of β-Pix with adhesions (Supplementary Fig. 22d). To examine how optoPlexin may affect β-Pix locally, we co-expressed mVenus-β-Pix or mCherry-Paxillin with optoPlexin in MC3T3-E1 cells and tracked their changes upon optoPlexin activation. Upon activating optoPlexin, β-Pix was rapidly depleted in illuminated region (Fig. 9a,b top panel, Supplementary Fig. 23a and Supplementary Movie 19). This depletion was not due to a loss of adhesions as we observed an increase in the intensity of paxillin around sites of illumination (Fig. 9c,d top panel, Supplementary Fig. 23b and Supplementary Movie 20). The opposite changes in β-Pix and paxillin accumulation indicate dissociation of β-Pix from adhesions in the illuminated area (Fig. 9e). The decrease of β-Pix around the region of illumination

preceded or coincided with the initiation of retraction (Fig. 9b top panel and Supplementary Fig. 23c). Moreover, we observed a concurrent accumulation of β-Pix and nascent adhesions in distal regions where new protrusions were produced (Fig. 9b,d bottom panels). In support of the involvement of RhoA signalling, optoPlexin-delPBD failed to deplete β-Pix locally and addition of ROCK inhibitor Y-27632 also abrogated the effects of optoPlexin on β-Pix (Fig. 9f and Supplementary Fig. 24). The presence of exogenous β-Pix caused a delay in retraction that was proportional to the level of β-Pix expression (Fig. 9g), suggesting that a local depletion of β-Pix may be required for the initiation of retraction. Taken together, these data revealed a novel mechanism in which a redistribution of β-Pix mediated by Plexin-B1-RhoA pathway coordinates CIL.

## Discussion

Semaphorin-Plexin signalling attracts a lot of research interest because of its important functions in development and diseases. *In vivo* semaphorins are often present as directional cues for cell migration, and thus experimental perturbations with spatial control at the subcellular scale can offer unique insight into their signalling mechanisms. One of the technical challenges in spatial control is that existing ligand-based methods may not be optimal for localized stimulation. Here we describe a novel approach to precisely control the location and time of Plexin-B1 activation with light. We validate this new optogenetic reagent by tracking its binding with two known interacting RhoGEFs, PRG and LARG, and by visualizing its activation of RhoA. We named the new tool optoPlexin, following the naming convention of many optogentic reagents developed in recent years[46–49]. To our knowledge, optoPlexin is the first optogenetic tool for the receptors of repulsive guidance molecules.

The optogenetic module Cry2 has two independent[50] modes of action, that is, inducible translocation mediated by CIB1 (ref. 31)

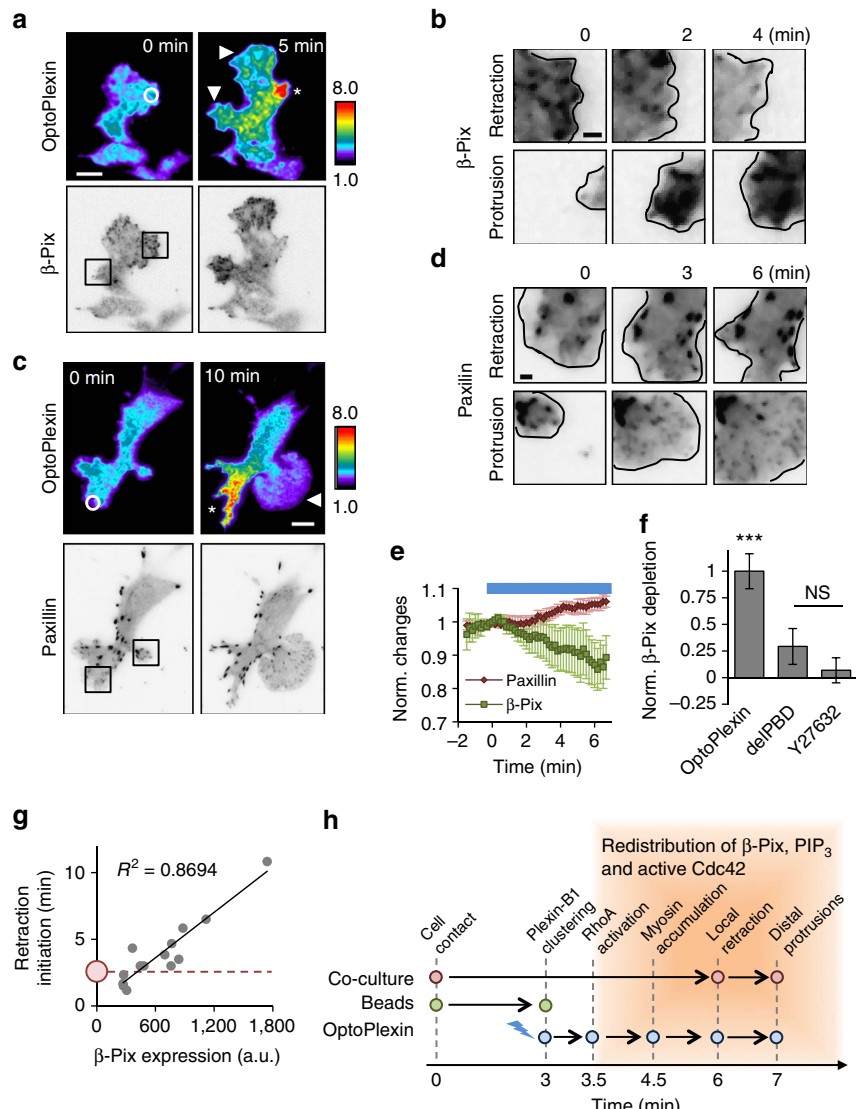

**Figure 9 | OptoPlexin mobilizes and redistributes β-Pix to distal regions.** TIRF images showing intensity changes of mCherry-optoPlexin and mVenus-β-Pix (**a**) or mVenus-optoPlexin and mCherry-Paxillin (**c**) in MC3T3-E1 cells upon local illumination. White circle, region of illumination. Scale bar, 10 μm. Asterisk, induced retraction. Arrowhead, induced protrusion. Inset squares in **a** and **c** are magnified to show temporal changes in mVenus-β-Pix (**b**) or mCherry-Paxillin (**d**) at induced retractions (**b,d**, top panels) or protrusions (**b,d**, bottom panels). Scale bar, 2 μm. (**e**) Quantification of fluorescence intensities of mCherry-Paxillin ($n = 10$ cells) and mVenus-β-Pix ($n = 18$ cells) in a 50 μm-diameter circle centred at the region of illumination on local optoPlexin activation in MC3T3-E1 cells. Blue line, illumination at 440 nm. (**f**) Quantification of mVenus-β-Pix depletion in a 50 μm-diameter circle centred at the region of illumination upon local illumination of MC3T3-E1 cells expressing optoPlexin, delPBD or optoPlexin with addition of 10 μM Y-27632 ($n = 10$–18 cells). (**g**) Delay in initiation of retraction upon local activation of mCherry-optoPlexin with respect to exogenous β-Pix expression levels in MC3T3-E1 cells Red circle indicates mean delay in initiation of retraction by local activation of mCherry-optoPlexin in MC3T3-E1 cells without exogenous β-Pix expression as measured in Fig. 3e. $R^2$, Pearson's correlation coefficient. (**h**) Model of Sema4D-Plexin-B1-mediated CIL between osteoclasts and osteoblasts. For **e,f**, means ± s.e.m. are shown. ***$P < 0.001$ and *$P < 0.05$, NS, not significant, Student's t-test.

and homo-oligomerization[32], which benefitted optoPlexin design to reduce dark background. We found that mere oligomerization (by omitting CIB-CAAX) is not sufficient for inducing binding of PRG, a GEF for RhoA (Fig. 3d,e) or RhoA activation (Fig. 3g). It is likely to be that additional factors such as Rnd1 or Rac1 on the plasma membrane may participate in recruiting PRG[20,51,52]. In addition, the slow off-rate of Cry2-CIB1 binding helped to maintain the Plexin-RhoGEF complex locally, mimicking sustained contact between osteoblasts and osteoclasts. Using optoPlexin, we demonstrated spatial modulation of different regulators of cell migration to understand the signalling mechanism involved in CIL between osteoclasts and osteoblasts.

As plexins share high sequence homology in their intracellular regions[53], optoPlexin design should be extendable to other plexins as well.

Upon Sema4D stimulation ErbB2 associates with and is transactivated by Plexin-B1 and regulates RhoA activation through phospholipase C-γ (PLC-γ)[37,38]. Given that the interaction with ErbB2 is mediated by the extracellular domain of Plexin-B1, it is unlikely to be that optoPlexin can associate with or activate ErbB2, which could be a limitation of optoPlexin. However, clustering of the cytosolic domain of Plexin-B1 on the membrane was shown to be sufficient for RhoA activation[17]. Our results also demonstrated that optoPlexin sufficiently recruits RhoGEF and activates RhoA.

Mutations designed to abrogate PLC-γ association (optoPlexin-YF) did not affect RhoGEF recruitment, RhoA activation or the repulsion phenotype induced by optoPlexin. It is possible that ErbB2-mediated tyrosine phosphorylation of Plexin-B1 and consequent association with PLC-γ promotes clustering of the endogenous protein, which is compensated by light induced clustering of optoPlexin.

In interrogating Plexin-B1 signalling, we probed the RhoA-ROCK-Myosin pathway and observed accumulation of MyoRLC and maturation of nascent adhesions in the protrusions where optoPlexin was activated. Local activation of optoPlexin repolarized active Cdc42 and active Rac1 away from the retracting area to new distal protrusions. We attribute these effects to mobilization and a redistribution of β-Pix. Cross-talk among Rho GTPases are known to facilitate cell migration and polarity[54]. β-Pix is one of the molecules that mediate cross-talk between RhoA and Cdc42 or Rac1. Myosin II-mediated contractility, downstream of RhoA/ROCK signalling, has been shown to induce dissociation of β-Pix from adhesions and decrease Rac1 activation[45]. Although we identified RhoA/ROCK activity being critical for local depletion of β-Pix, the exact molecular mechanism by which β-Pix is regulated demands further investigation. Our observations of opposite repolarization of MyoRLC and adhesion maturation compared to β-Pix are consistent with a model of mechanical regulation, although we cannot rule out a role of ROCK-mediated phosphorylation of β-Pix[43,55] in its redistribution. Importantly, localized Plexin-B1 signalling did not simply deplete β-Pix, but induced redistribution to distal regions and activated Cdc42 and Rac1 to promote new protrusions. Although we also observed repolarization of PIP_3, the underlying signalling mechanism remains to be explored. Plexin-B1 may inhibit PIP_3 by activating phosphatase and tensin homologue (PTEN) via its RasGAP domain[56]. RhoA activation by Plexin-B1 may also activate PTEN and SH2 domain containing inositol 5′-phosphatase 2 (SHIP2) (refs 57,58). In other studies, Plexin-B1 has been reported to promote PIP_3 production through activation of phosphoinositide 3-kinases[59,60]. R-Ras is another important target that regulates CIL, as evident from the inability of optoPlexin-RA to induce CIL or PIP_3 redistribution. All plexins contain a RasGAP domain[16,54] and other repulsive guidance molecules such as ephrins also inhibit R-Ras[61,62], suggesting that R-Ras may play a critical role in mediating repulsion between cells.

Optogentic approaches offer spatial and temporal control in initiating signalling pathways in subcellular regions. This is particularly lucrative in studying CIL, as the underlying signalling is initiated specifically at the site of cell–cell contact. Precise activation of optoPlexin in leading protrusions steered MC3T3-E1 cells away in a manner reminiscent of osteoclast-mediated CIL and followed similar timelines of retractions and protrusions (Fig. 9h). Although CIL has been proposed to transiently promote cell motility[63], both osteoclasts and localized optoPlexin stimulation induced CIL without affecting the overall motility of MC3T3-E1 cells (Figs 2b and 5c, and Supplementary Fig. 17). Previous studies have reported formation of transient cadherin-mediated cell–cell adhesions that generate mechanical tension across cells undergoing CIL[7–9,64,65]. We observed a sustained contact between osteoclasts and osteoblastic cells before separation (Supplementary Movies 2, 3 and 5), which may potentially be critical for Plexin-B1 clustering. Whether intercellular adhesions form between osteoblasts and osteoclasts, and whether they affect heterotypic CIL remain to be investigated. Our experiments with optoPlexin cannot decipher the nature and duration of such cell–cell contact. Nevertheless, we observed cellular response comparable to CIL induced by osteoclasts with similar kinetics (Fig. 9h), suggesting that localized activation of Plexin-B1 was sufficient to generate similar behaviour. Furthermore, our observations on spatial redistribution of PIP_3,

active Cdc42 and active Rac1 suggest that although different signalling mechanisms may initiate homotypic and heterotypic CIL, their effects converge to induce similar spatial changes in factors that regulate cell migration.

In recent years, repulsion between cells has been implicated in many physiological and pathological processes[1]. Our data demonstrate CIL between osteoblasts and osteoclasts. In the bone microenvironment, resorption by osteoclasts is followed by deposition of matrix by osteoblasts. Osteoclasts are generally thought to attract osteoblasts to the site of resorption through release of chemoattractants[24–27]. However, a simple chemotaxis model does not explain spatial segregation of osteoclasts and osteoblasts on the bone surface[28], which is consistent with their opposing functions. Our observation of CIL between osteoclasts and osteoblasts explains the spatial segregation between osteoblasts and osteoclasts[6,28]. In addition, loss of Sema4D has been reported to impact bone resorption[29]. It would be interesting to investigate whether osteoclasts utilize similar mechanisms to repel osteoblastic bone lining cells, thereby gaining access to the bone matrix to initiate resorption. A combination of CIL and chemotaxis might facilitate osteoblasts occupying sites of resorption only after the short-lived osteoclasts disappear. Considering the multitude of coupling factors between osteoclasts and osteoblasts, further research is necessary to gain a complete understanding of how osteoclasts modulate osteoblast migration and function.

## Methods

**Cell culture and transfection.** POB were obtained as described previously[66] from two month old inbred C57BL/6 male mice. Briefly, calvariae from five to six neonatal mice were minced, washed with PBS and digested with 0.5 mg ml$^{-1}$ collagenase P (Roche Diagnostics, Indianapolis, IN) and 0.5 mg ml$^{-1}$ Trypsin/EDTA in PBS at 37 °C four times for 10 min each and a final digest for 90 min. Cells isolated from digests 2–5 were combined and plated at a cell density of $4 \times 10^4$ cells per well in six-well dishes for imaging experiments. For coculture experiments, POBs were lifted at day 5–7 after isolation. For real-time quantitative PCR, MC3T3-E1 and POBs were plated at $5 \times 10^4$ cells per well in six-well dishes and cultured in osteoblast differentiation media for 7 or 14 days. Osteoblast differentiation medium consisted of basic medium plus 50 µg ml$^{-1}$ phosphoascorbate (Wako Pure Chemical Industry, Osaka, Japan). Media were changed every 3 days. β-Glycerophosphate (5 mM; Sigma-Aldrich, Saint-Louis, MO) was added on day 7 of these cultures.

For osteoclastic differentiation, recombinant mouse macrophage-colony stimulating factor (M-CSF) and RANKL were purchased from R&D Systems (Minneapolis, MN, USA). BMMs were made as described previously[66] from five to six 2-month-old inbred C57BL/6 and outbred CD1 male mice. Briefly, mice long bones were dissected and the bone marrow was flushed out. The cells were suspended in growth medium and plated in petri dishes with 100 ng ml$^{-1}$ M-CSF and grown for 2–3 days. Adherent cells after 2–3 days were used as BMMs for further experiments. BMMs were plated at $6 \times 10^4$ cells per well in 12-well dishes in basic medium plus 30 ng ml$^{-1}$ M-CSF with or without RANKL (30 ng ml$^{-1}$) for 3 days. Vehicle for RANKL and M-CSF was 0.1% BSA in PBS.

All studies were conducted in accordance to the approved protocols by the Institutional Animal Care and Use Committee of the University of Connecticut Health Center.

COS-7 and MC3T3-E1 cells were obtained from American Type Culture Collection and were cultured in DMEM and α-MEM (Lonza, Basel, Switzerland) basal media, respectively, and were passaged every third day of culture. For optimal growth, the media were supplemented with 10% (v/v) fetal bovine serum (Gibco, Billings, MO) and Penicillin/Streptomycin (Lonza), and the cells were maintained under standard cell culture conditions (37 °C and 5% CO$_2$). The cell lines were regularly checked for mycoplasma contamination. FuGENE 6 reagent (Promega, Madison, WI) was used for transient transfections according to manufacturer's instructions. Lentiviral transductions for shRNA and CRISPR-Cas9 approaches were performed as previously described.

**DNA plasmids.** pLKO.1 puro vector (Addgene plasmid 8453) was a gift from Dr Bob Weinberg. LentiGuide-Puro (Addgene plasmid 52963) and lentiCas9-Blast (Addgene plasmid 52962) were gifts from Dr Feng Zhang (Massachusetts Institute of Technology, Cambridge). The farnesylated CIB (CIB-CAAX) and full-length Cry2 expression plasmids (same as addgene plasmids 28240 and 26871, respectively) were gifts from Dr Chandra Tucker (University of Colorado, Denver). The source complementary DNAs of mouse Plexin-B1, Plexin-B1-RA and Plexin-B1-delPBD (Supplementary Fig. S1) were from Dr Hiroshi Takayanagi

(Tokyo Medical and Dental University, Japan). The cDNAs of PRG and LARG were from Dr Keith Burridge (University of North Carolina at Chapel Hill), β-pix from Dr Peter Hordijk (University of Amsterdam, the Netherlands), myosin regulatory light chain from Dr Rex Chisholm (Northwestern University) and PH-Akt from Dr Craig Montel (University of California, Santa Barbara). These constructs were initially subcloned into the pTriEx-4 vector (Novagen) using PCR and restriction digestion. Cry2 was positioned at the N terminus and Plexin-B1 at the C terminus to minimize interference with PRG and LARG. As indicated in the results and figure legends, tags of compatible fluorescent proteins including mCerulean, mVenus and mCherry were appended to facilitate detection. Unless specified otherwise, the termini of tagging were positioned as in the orders they were written. Additional point mutations in the Cry2, D387A and D393A were generated using overlapping PCR. The open reading frames of all DNA plasmids were verified by DNA sequencing.

**Co-culture migration assay.** Cell culture inserts (Ibidi, Germany) were used according to the manufacturer's instructions. Briefly, the culture inserts were placed on tissue culture dishes and BMMs were cultured and differentiated into osteoclasts inside one of the insert chambers. Upon appearance of multinucleated osteoclasts, osteoblasts or MC3T3-E1 cells were added to the empty adjacent chamber. After 5–6 h of incubation allowing the cells to adhere to the tissue culture dish, the inserts were gently lifted using tweezers enabling the cells to migrate towards the osteoclasts. Delay in retraction and protrusions upon contact were determined by visual inspection. Initiation of retraction was identified as the time point when a protrusion in contact with osteoclasts began to contract. Initiation of protrusions were identified as the time point of formation of the persistent protrusions that enabled the cell to migrate away from the site of contact. Cell velocities were computed by measuring the displacement of the cell nuclei.

**Quantification of CIL.** Whether a protrusion collapsed at the site of cell–cell contact was determined by visual inspection using time-lapse images within 30 min after contact. Previous studies have quantified CIL by measuring spatial separation between cells undergoing CIL[7,9,67]. As osteoclasts are large cells with multiple nuclei, we determined the spatial separation between osteoclasts and osteoblasts by computing the difference in distances between the osteoblastic nucleus and the site of contact at the time of and 40 min after contact. Positive and negative values respectively indicate spatial separation or lack of it. Changes in direction of migration of osteoblasts after contact were assessed by measuring Cx using vector analyses as described previously[11,67,68]. Briefly, the displacement of a migrating osteoblast for 40 min before contact (vector A) and 40 min after contact (vector B) were determined by tracking the nuclei. The Cx component of the vector (B–A) is a measure of how much the cell has deviated from its original trajectory (vector A′) in the migration axis after contact (Fig. 1h). Cx values for MC3T3-E1 cells that did not come in contact with osteoclasts were also determined for 40 min intervals. Positive and negative values of Cx respectively indicate persistence and reversal of direction of motion. In the absence of CIL, Cx approaches zero, whereas higher magnitudes indicate significant deviation from expected path.

**Real-time quantitative PCR.** Total RNA was extracted using Trizol (Invitrogen) following the manufacturer's instructions. Four micrograms of total RNA was DNase treated (Ambion Inc., Austin, TX) and converted to cDNA by the High Capacity cDNA Archive Kit (Applied Biosystems, Foster City, CA). PCR was performed in 96-well plates. Assays-on-Demand Gene Expression Taqman primers (Applied Biosystems) were used for PCR (Mm99999915_g1 for *Gapdh*; Mm00443147_m1 for *Sema4d*; Mm00555359_m1 for *Plxnb1*). *Gapdh* served as endogenous control. All primers were checked for equal efficiency over a range of target gene concentrations. Each sample was amplified in duplicate. PCR reaction mixture was run in Applied Biosystems Prism 7300 Sequence Detection System instrument utilizing universal thermal cycling parameters. Data analysis was done using relative quantification ($\Delta\Delta C_t$) or the relative standard curve method. As per the manufacturer's recommendation, any cycle threshold ($C_t$) values obtained below 33 were considered undetectable for that particular target gene.

**Western blot analysis.** MC3T3-E1 and POBs prior and after addition of β-glycerophosphate were washed three times with PBS and lysed on ice in a buffer containing 1% Triton X-100, 50 mM monobasic sodium phosphate (pH 7.4), 150 mM NaCl, 5 mM EDTA and Halt Protease inhibitor cocktail (Thermo Scientific, Rockford, IL, USA). The cell lysates were fractionated using 4–12% NuPAGE gels (Invitrogen), immobilized on to polyvinylidene difluoride membranes (Millipore, Germany), followed by immunoblotting using anti-Plexin-B1 monoclonal antibody (A-8, Santa Cruz Biotechnology, Dallas, TX, USA) or anti-α-tubulin monoclonal antibody (DM1A, Cedarlane Laboratories, Burlington, Ontario, Canada) and horseradish peroxidase-coupled secondary antibody (Millipore). Enhanced chemiluminescence detection (Pierce, Waltham, MA, USA) and autoradiography were used to detect the signals. For detection of Sema4D-Fc horseradish peroxidase-coupled Protein-A was used. The blot images were cropped to show relevant bands. For Fig. 1c, the uncropped scan is shown in Supplementary Fig. 25.

**RNAi approaches and CRISPR-Cas9-mediated Plexin-B1 knockout.** siGEN-OME non-targeting (D-001206-13-05) and Plexin-B1 (M-040982-01-0005) siRNA pools (GE Dharmacon, Colorado, USA) were used according to manufacturer's instructions.

shRNA target sequences for luciferase (Luc, TRCN0000072254) and Plexin-B1 (sh1—TRCN0000078913, sh2—TRCN0000078917 and sh3—TRCN0000078916) were identified using Broad Institute Genetic Perturbation Platform web portal (http://broadinstitute.org) and were cloned into pLKO.1 puro vector.

To identify target sites for CRSPR-Cas9-mediated knockout, the genetic sequence of Plexin-B1 was obtained from UCSC genome browser (http://genome.ucsc.edu) using the mouse assembly GRCm38/mm10 (December, 2011). The gene encoding Plexin-B1 has 38 exons, with the start codon located within the third exon. Two single-guide RNAs (sgRNAs) were designed flanking the start codon (Supplementary Fig. 3f and Supplementary Table 1) using CRISPR design tool by Zhang lab (http://crispr.mit.edu) so that Cas9-mediated cleavage was expected to delete a 100 bp region including the start codon. The sgRNAs were cloned into lentiGuide-Puro vector[69]. Cas9 and the sgRNAs were stably expressed in MC3T3-E1 cells using lentiviral transduction. Genomic DNA from 24 clones of the transduced MC3T3-E1 cells were extracted using Quick-gDNA kit (Zymo Research, Irvine, CA, USA) as per the manufacturer's instructions. Upon PCR screening we identified two clones that showed deletion at the expected site (Supplementary Fig. 3g and Supplementary Table 1). DNA sequencing validated the loss of Plexin-B1 start codon in both of these clones. Loss of Plexin-B1 expression was further confirmed using western blotting (Fig. 1d).

**Immobilization of Sema4D-Fc and IgG1 on silica beads.** Sema4D-Fc and hIgG1 were immobilized on 5 μm diameter Silica Bind-IT pre-activated microspheres (catalogue number SB06N) from Bangs Laboratories (Fishers, IN, USA) by following the manufacturer's instructions with minor modifications. Briefly, 50 μl of the 2.5% bead suspension was centrifuged at 1,000 g for 5 min. The supernatant was removed and the beads were washed thrice in 100 μl coupling buffer (50 mM MES buffer, pH 5.2). After the final wash, the supernatant was removed and 25 μl coupling buffer. Twenty-five microlitres of Sema4D or hIgG1 solution (0.1 μg ml$^{-1}$) were added to the bead suspension and incubated for 2 h at 4 °C for 2 h on a rotator. The suspensions were centrifuged, the supernatant was removed and the beads were washed thrice in 100 μl storage solution (150 mM NaCl, pH 7.0). After the final wash, the beads were resuspended in 50 μl storage solution and this suspension was used for further experiments. To assess the amount of Sema4D immobilized on the beads, western blotting was performed under non-reducing conditions. The samples $1\times$ and $1/10\times$ dilutions were equivalent to 25 and 2.5 μl of the bead suspension. From the western blotting (Supplementary Fig. 9d,e) we estimated that 80.5 ng Sema4D-Fc was present in the $1/10\times$ sample. The density of the beads as provided by the manufacturer is 2 g cm$^{-3}$. Assuming no loss of beads or Sema4D-Fc during the sample preparation and assuming perfectly spherical beads, we calculated that the average number of Sema4D-Fc molecules bound to each silica bead was around 1,000. Assuming a hemispheric contact (Supplementary Fig. 9f), each bead would provide 500 Sema4D-Fc molecules to come in contact with the cells. Furthermore, an even distribution of Sema4D-Fc is equivalent to a surface density of 12.73 molecules per μm$^2$, which amounts to a distance of 280 nm between each molecule (Supplementary Fig. 9g). It is possible that by using a significantly higher surface density of Sema4D-Fc, or enabling diffusion of Sema4D-Fc on such beads by using lipid layers may solve this problem; however such approaches would still not allow sufficient spatial and temporal control since it is difficult to manipulate the specific time and location of contact between cells and the beads.

**Imaging setup.** All time-lapse imaging were performed on a customized Nikon Ti-E inverted microscope. Phase-contrast images for osteoclast co-culture experiments were captured using a $10\times$ objective (numerical aperture (NA) 0.30) and Andor Neo 5.5 sCMOS camera. Wide-field fluorescence imaging was performed using $\times40$ oil objective (NA 1.30) using LED sources for excitations at 438, 513 and 575 nm (Lumencor) for imaging mCerulean, mVenus and mCherry, respectively. TIRF imaging was performed using a $\times60$ oil TIRF objective (NA 1.49). The microscope was modified with a 'stage-up' design, which enables an insertion of two independent, motorized dichroic mirrors/filter cubes in the microscope infinity space. A dichroic mirror in the bottom cube was used to reflect excitation laser lines at 442, 514 and 594 nm for imaging of mCerulean, mVenus and mCherry, respectively. The laser lines were lunched from a fiber-coupled LMM5 system (Andor) equipped with an acousto-optic tunable filter for shutter and intensity control. Another dichroic mirror (495LP) in the top cube was used to bring in optogenetic illumination originated from a LED source at 440 nm (CoolLED). The 495LP mirror permitted immediate acquisition of mVenus or mCherry after optogenetic illumination. Alternatively, the top mirror can be rotated out to a blank position for Patterned illumination was generated using a commercial digital mirror device (Mosaic, Andor). The fluorescent emission was captured with an electron-multiplying charge-coupled device camera (iXon Ultra, Andor). Metamorph software was used to control the imaging set up. Live cell imaging was performed at 37 °C in a heated chamber (Bioptechs) with humidified 5% CO$_2$ supply. Vitamin and phenol red-free media (US Biological) supplemented

with 2% fetal bovine serum were used in imaging to reduce background and photobleaching.

**Membrane recruitment assay.** COS-7 or MC3T3-E1 cells were transiently transfected with mVenus- or mCherry-optoPlexin along with CIB-CAAX. The recruitment of optoPlexin to the plasma membrane was induced with a short pulse (50 ms) 440 nm illumination at 10 s intervals, shortly (typically 10–15 frames) after acquisition of base lines of fluorescent intensities. For localized activation, a 5 μm-diameter circular region near the cell periphery was chosen for each cell. TIRF imaging of mVenus and/or mCherry channel was used to gauge the amount of optoPlexin that became excitable by the evanescent wave (∼200 nm from the coverslip surface) presumably due to association with the plasma membrane. As a robust translocation of optoPlexin could be induced in the assay (Fig. 1c), proteins associated with optoPlexin, labelled with different fluorescent proteins for example, mVenus-PRG and mCherry-optoPlexin, for example, may also 'precipitate' into the TIRF imaging plane due to association. After conducting background subtraction using the mean fluorescence intensities of cell-free regions, the fluorescence intensities after ten blue light pulses were normalized to that before blue light illumination and the changes were used to estimate the recruitment of optoPlexin and its associated proteins to the plasma membrane. The results of membrane recruitment assay were quantified as follows,

$$R_C = \frac{\Delta F/F(mVenus)}{\Delta F/F(mCherry)}$$

where $R_C$ = relative co-recruitment and $F$ = fluorescence intensity.

To minimize effects of noise in measurement of binding, only cells with at least 30% fractional increase in optoPlexin signal were considered.

**Quantification of optoPlexin clustering.** The local variance of fluorescence intensities was used to estimate the extent of molecular clustering of optoPlexin and its associated proteins. Briefly, TIRF imaging was used to sample the basal focal plane of cell, thereby rejecting most of the out-of-focus fluorescence. The raw images were first background subtracted using cell-free regions. To eliminate effects of changes in cell topology on changes in variance of fluorescence intensities, a ratio image was then calculated using the original image divided by a processed one, which was passed through an empirically determined $5 \times 5$ median filter. Clustering index was defined as the fractional changes in s.d. of the ratiometric image (Supplementary Fig. 13).

**Analysis of cell repolarization.** Fluorescence images of MC3T3-E1 cells expressing optoPlexin (or its mutants) were thresholded based on intensity to produce binary images. Cell velocity at each frame was measured by tracking the displacement of the centroids of these binary images and average of such velocities was used to assess the cell velocity. Changes in cell shape were identified by subtracting the binary images in a time series from the image at time 0. Pixels of areas in positive values were identified as protrusions. Conversely, ones in negative values were identified as retractions. Areas where the images overlapped since time 0 were designated as neutral regions. An arbitrary floor was added to the image to display these regions in colour: red for protrusion, blue for retractions and green for regions occupied by the cell at both time points, whereas cell free regions were shown in white. The region of blue light illumination is shown in yellow (Fig. 5d). Only regions exceeding 50 pixels (3.52 μm²) were considered in the quantification of protrusions and retractions, and only regions that were converted from protrusions to retractions were used to measure induction of retractions by optoPlexin.

To measure the angle of retraction ($\theta_1$) or protrusion ($\theta_2$) with respect to the region of illumination, a straight line joining the centroid of the cell at time 0 was used as the horizontal axis. The angle between the straight line joining the centroid of protrusions (or retractions) and the centroid of the cell at time 0 and the horizontal axis was defined as the angle of protrusion (or retraction). The proximity of retraction ($d_1$) and protrusion ($d_2$) to the region of illumination was measured by the linear distance between the centroid of protrusion (or retraction) and the centroid of the region of illumination. Delay in initiation of retraction was determined by identifying the time point when the cell area in a 50 μm-diameter circle centred around the region of illumination (as defined for Fig. 5d) began to contract. Metamorph (Molecular Devices, Sunnyvale, CA) and Matlab (Mathworks, Natick, MA) softwares were used for image processing and data analyses.

**Imaging and analysis of FRET sensors.** For FRET imaging, donor (mCerulean) images were captured sequentially after the FRET (mVenus) image. The Dora-RhoA sensor uses an intra-molecular effector fragment derived from the RhoA effector protein kinase N to detect the nucleotide states (GDP or GTP) of RhoA. Upon GTP-loading or activation of RhoA, Dora-RhoA interacts with the effector fragment and adopts a closed conformation. The conformational change leads to an increase in FRET signal, the extent of which is quantifiable using the ratio of FRET/CFP. For RhoA sensor imaging in MC3T3-E1 cells upon Sema4D-Fc or hIgG1 treatmentm mCerulean and FRET images were acquired every minute.

For RhoA sensor imaging in COS-7 cells on optoPlexin activation, sequential images of mCherry, mCerulean and FRET images were acquired every 10 s using TIRF excitation with 594 and 442 nm lasers. As the excitation wavelength for CFP and FRET was sufficient to activate Cry2 recruitment, no additional blue light illumination was employed. Ratiometric images of FRET and mCerulean were generated and changes in the average intensity of these ratiometric images were used to measure changes in RhoA activity. The first image in each of these time series was used for baseline measurement of RhoA activity before optoPlexin activation. Changes in ratiometric measurements for a binding-deficient RhoA control sensor in response to mCherry-tagged Cry2 recruitment was used to correct for optical artefacts of mCherry and Cry2 recruitment to the TIRF plane.

DORA-Rac1 and DORA-Cdc42 sensors share a similar design as the Dora-RhoA sensor, except using a Cdc42/Rac interactive binding domain of p21-activated kinase as the binding domain. The non-binding control Cdc42 sensor control harbours a mutated (H83,86D) effector domain in Dora-Cdc42 and does not alter FRET in response to activation. For Cdc42 sensor, Cdc42 control sensor and Rac1 sensor imaging in MC3T3-E1 cells, optoPlexin was locally activated using the Mosaic illumination system as mentioned above until we observed local retraction and distal protrusions, at which point the time lapse imaging was ended and FRET and mCerulean images were obtained immediately afterwards. Ratiometric images from each of these FRET and mCerulean image pairs were processed as described previously[70] and used to visualize the spatial distribution of Rac1 and Cdc42 activities in individual cells.

**Statistical analyses.** Microsoft Excel and Matlab were used for statistical analyses. Sample sizes for different experiments were chosen based on the commonly used range in the field without conducting any statistical power analysis. Normal probability plot function of Matlab was utilized to confirm normal distribution of the data. Similarity of variance across all the groups (or before and after stimulation) being compared were tested using F-test in Matlab. For reverse transcriptase–PCR data, one-way analysis of variance was performed using Sigmaplot 11.0 (Systat Inc., Chicago, IL). Sample means and s.e.m. was calculated and shown on the graphs. P-values were obtained from one-tailed Students t-test.

**Data availability.** All crucial data supporting the findings of the study are included in the article and the Supplementary Information. Additional raw data are available from the corresponding author upon request.

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

## Acknowledgements

We thank Dr A. Cowan for critical reading of the manuscript, and C. Sirois and Dr S. Chamberlain for help with CRISPR-Cas9-mediated genome editing. This study was supported by N.I.H. grants GM117061 (Y.I.W.), GM62290 (V.R.) and AR060286 (C.P.).

## Author contributions

A.D.R. initiated the project, and designed and performed most of the experiments and data analyses. T.Y. developed the DORA biosensors. S.C. characterized Sema4D and Plexin-B1 mRNA expressions, isolated BMMs and POBs, and differentiated osteoclasts for the coculture assays. Y.I.W. guided the project and wrote the final version of the manuscript based on contributions from all the authors.

## Additional information

**Competing interests:** The authors declare no competing financial interests.

