## [Peer Review File · Nature Communications]

Reviewers' Comments:

Reviewer #1 (Remarks to the Author)

In this manuscript the authors develop a new optogenetic approach to control the activity of Plexin. They find that optogenetic activation of Plexin leads to repolarization of small GTPases, Myosin II and PIP3. Time lapse microscopy shows that protrusion are collapsed in the region where Plexin is activated, whereas new protrusion are formed away from the site of Plexin activation. The authors interpret these observation as Plexin being involved in Contact Inhibition of Locomotion (CIL) and based on their results they propose a signalling cascade downstream of Plexin as playing a role in CIL.

This is an interesting and well presented study that makes important contributions to the field of cell migration. One aspect of this study deals with the development of a new optogenetic tool to study Semaphorin/Plexin signalling in a localized manner. This in itself is an important achievement. A second facet of this work is the role that semaphoring/Plexin could have in CIL. Some of the results concerning this second part are intriguing but additional experiments need to be done. In particular the analysis of a CIL response need to be better characterized.

Specific comments:

1. One of the conclusions of this work is that Semaphorin/Plexin is involved in CIL of osteoblasts (and MC3T3-E1 cells). The authors need to demonstrate this directly by analysis collision between these cells to determine whether these cells exhibit a CIL response. This is an important experiment as not all cell display CIL. Once they have determined that the cells that they are using in their study exhibit CIL, they need to show that this CIL response is dependent on Semaphorin/CIL. Loos of function experiment of these molecules would be the most straight forward approach. Without this basics characterization of CIL the author cannot describe the response trigger by Plexin activation as CIL.
2. The characterization of CIL presented in this study is not adequate. CIL involves cell migration and not only collapse of cell protrusions. The authors should perform longer movies to see whether localized activation of Plexin leads to a change in the direction of cell migration, as expected if CIL is involved. Velocities before and after cell collision and acceleration should be determined. In addition, these changes in cell migration should be compared with the changes induced by proper cell-cell collision (point 1). The data (and movies) presented are too short to see a proper change in the direction of migration, as it is only possible to appreciate a change in the orientation of protrusions.
3. The discussion is rather poor and there are aspects of CIL that do not completely fit with the data presented here. For example, it has been recently shown that CIL involves generation of tension across the cell-cell contact, and this could be an important aspect of the CIL response (Davis et al., 2015; Cell 161, 361). However, the "CIL response" induced in this manuscript cannot involve tension across the cell-cell contact. In addition, it has been shown that during CIL the protrusions away from the cell contact are formed before cell-cell separation (Scarpa et al., 2015; Dev Cell 34, 421), but in this manuscript the authors show that the protrusions away of the cell contact are produced after cell contraction in the region that would correspond to the cell-contact (activation of Plexin). The authors need at least discuss their findings in relation to what is known about CIL.
4. In Fig 2f, the levels of Plexin activation do not coincide with the levels of RhoA activity: protrusions at the right-bottom corner exhibit low levels of Plexin and high levels of RhoA activity.

This is not consistent with the hypothesis that activation of Plexin leads to an immediate activation of RhoA.

Reviewer #2 (Remarks to the Author)

Optogenetics has been increasingly emerging as a powerful approach for basic investigation of cellular signaling, particularly in situations where spatial and temporal variation of signaling plays an important role in cell behavior. This manuscript utilizes Optogenetics to study cell protrusion and migration in response to semaphorin-plexin signaling. Specifically, investigators leverage prior findings in the field that Cry2 can both heterodimerize (with CIB1) and homodimerize in response to blue light, by fusing the endodomain of plexins to Cry2 and tethering CIB1 to the cellular membrane. Illumination induces plexin membrane localization and oligomerization, leading to a range of downstream responses including RhoA activation, cell process retraction, process protrusion at a distal site, and redistribution of Rac1/Cdc42/beta-Pix from the retracting to the protruding sites. The placement of a plexin under light control is novel, the experiments are well-designed, the data are high in quality, and the suggestions of downstream mechanism are intriguing. There are only several questions.

If the intracellular domain of a signaling receptor is oligomerized, it is not entirely clear that membrane localization would be required for activation. For example, in other systems involving optogenetic activation of receptors (e.g. LRP6) activation can occur without membrane localization. Can the authors speculate why membrane localization would be required for signal activation in this system? Are the effectors investigated in this study (e.g. GEFs) also membrane-localized?

For optoPlexin expressing cells, the average time between signal activation and process retraction was 2.5 minutes. How does (and other aspects of downstream signal activation) quantitatively compare to the local administration of the Sema ligand? Clearly optogenetic activation affords broader control of pathway activation as a function of space and time, but it would help to benchmark relative to the natural signal a bit more.

While Figure 3 shows coincident retraction and protrusion in a number of cells, it is unclear what fraction of cells with focally illuminated and retracting processes had a subsequent protrusion. Was it 100%?

The relationship between beta-Pix, RhoA, Rac1, Cdc42, and plexin is unclear. There are a number of studies indicating that RhoA, Rac1, and Cdc42 can antagonize one another, such that RhoA activation alone could be hypothesized to contribute to the deactivation of Rac1 and Cdc42. Rather than examine this possibility, the investigators investigate beta-Pix, which clearly does exhibit an interesting depletion in illuminated regions, which could contribute to the coincident depletion of Rac1/Cdc42 activity and localization in such regions. However, it is unclear what causes beta-Pix to become depleted in response to plexin activation. Can the authors comment further on this, as well as the relationship between RhoA and Rac1/Cdc42 activity?

In sum, the optoPlexin design and subsequent logical sequence of experiments to begin to apply this tool to study downstream signaling is elegant. Addressing several questions, and as a byproduct adding more depth to the Discussion, would benefit the manuscript.

Reviewer #3 (Remarks to the Author)

This article reports the development of a photo-activation system for the cell surface receptor PlexinB1. Results show that upon blue light illumination, the optoPlexin reagent is recruited at the plasma membrane together with the RhoGEFs PRG and LARG, which interact with the C-terminal PDZ binding motif of PlexinB1. FRET experiments revealed that activity of optoPlexin mediates activation of RhoA, a known downstream effector of PlexinB1. The authors then used this system

to study the effect of local activation of PlexinB1 in a murine osteoblastic cell line (MC3T3-E1). They observed a cellular response that is reminiscent of contact inhibition of locomotion (CIL) - with retraction of the illuminated protrusion and extension of a new protrusion away from the site of illumination. This response is dependent on the RhoA/ROCK signaling, known to negatively regulate protrusion formation, but also involves the GAP activity of PlexinB1. Concomitant to retraction, the authors observed a repolarization of PIP3, cdc42, Rac1 and the RhoGEF beta-pix at the new protruding front.

Overall the data are interesting and convincing, although I feel that the behavior of cells that follows photo-activation of optoPlexin could have been more extensively documented. For example, the statement that « morphological changes altered the migration direction and caused the cell to migrate away from the site of illumination » (line 223-224) is not demonstrated (Fig.4 only shows that cell velocity remains unchanged).

One may wonder, however, how relevant these findings are for the biology of osteoblasts. To what extent does photo-activation of optoPlexin mimic an interaction with Sema4D-expressing osteoclasts? The authors should provide evidence that heterotypic CIL can occur between osteoblasts and osteoclasts and that it is mediated by Sema4D/PlexinB1 signaling. It is important because a role of the Sema4D/PlexinB1/RhoA/ROCK signaling axis has been previously reported by Negishi-Koga et al. to stimulate motility of osteoblastic cells. How do the authors reconcile this result with their findings of local retraction/collapse after optoPlexin activation? What I worry about is that important co-receptors for PlexinB1 function (such as ErbB2) may not be recruited and/or trans-activated by the optoPlexin construct. Therefore, the biological outcome of optoPlexin signaling would have little physiological relevance, at least in the context of the study of osteoblastic cells.

Another major concern that should be addressed is that, despite evidence for a spatial redistribution of several molecules contributing to polarized cell migration (cdc42, Rac1, Pip3, beta-PIX), this study does not provide functional evidence for their implication in PlexinB1-mediated CIL, nor on how PlexinB1 signaling may be mechanistically coupled to this polarity switch.

Minor points:

Control experiments showing membrane recruitment of optoPlexinRA are missing.

The article by Polleux et al. reports a role of Sema3A gradient on the patterning of apical dendrites of cortical pyramidal neurons, not on "outward radial migration of neurons in a developing cortex" (line 48-49)).

Line 46 writes "Semaphorins from family 3 and 7 are secreted (...)". Actually, Sema7a is not secreted but is a glycosylphosphatidylinositol (GPI)-anchored membrane glycoprotein.

The figure legends for Figures 8f and 8g are swapped (p.42).

Reviewer #4 (Remarks to the Author)

The authors use an optogenetic approach to achieve localised activation of a truncated Plexin B1 construct. They show that localised activation of Plexin B1 causes localised cell retraction. Given that global activation of the Plexin is known to give global retraction, this is somewhat predictable.

The authors describe the localised optical activation of Plexin B1 as contact inhibition (CIL). While these assays may mimic CIL, they are not replicating the conditions of CIL. The truncated Plexin has no transmembrane domain and does not see an immobilised semaphorin on an adjacent cell. It is difficult to see what advantage the optogenetic approach gives here, compared with looking at actual CIL with another cell.

The authors extend their imaging based analysis to analyse Rho GTPase signalling downstream of localised Plexin activation. They make some useful observations about localised Rho signalling; however, these are either based on known Plexin signalling or broadly predictable from the wider literature on cytoskeletal signalling.

Overall, the study is well done and the experimental work is of a good standard. I think that this is publishable work; however, I do not think it meets the criteria for publication at this level. My major criticism is that the optogenetic approach requires compromises without giving benefits. There is no reason not to study actual CIL with a full length Plexin. Secondly, although the signalling work is valuable, the advance is not enough for publication at this level.

Reviewer #1

In this manuscript the authors develop a new optogenetic approach to control the activity of Plexin. They find that optogenetic activation of Plexin leads to repolarization of small GTPases, Myosin II and PIP3. Time lapse microscopy shows that protrusions are collapsed in the region where Plexin is activated, whereas new protrusion are formed away from the site of Plexin activation. The authors interpret these observation as Plexin being involved in Contact Inhibition of Locomotion (CIL) and based on their results they propose a signalling cascade downstream of Plexin as playing a role in CIL.

This is an interesting and well presented study that makes important contributions to the field of cell migration. One aspect of this study deals with the development of a new optogenetic tool to study Semaphorin/Plexin signalling in a localized manner. This in itself is an important achievement. A second facet of this work is the role that semaphoring/Plexin could have in CIL. Some of the results concerning this second part are intriguing but additional experiments need to be done. In particular the analysis of a CIL response need to be better characterized.

We appreciate the reviewer's generous comments on the overall study and recognition of our efforts in developing the new optogenetic tool for Plexin-B1. We also completely agree with the reviewer that, in our initial submission, convincing evidence on the role of endogenous semaphorin/plexin in CIL was lacking. As suggested by the reviewer, in the revised manuscript we have provided new data on the role of endogenous Plexin-B1 in CIL in osteoblasts by conducting loss-of-function studies and offered better

characterization of CIL as detailed in Specific Comment 1 and 2.

Specific comments:

1. One of the conclusions of this work is that Semaphorin/Plexin is involved in CIL of osteoblasts (and MC3T3-E1 cells). The authors need to demonstrate this directly by analysis collision between these cells to determine whether these cells exhibit a CIL response. This is an important experiment as not all cell display CIL. Once they have determined that the cells that they are using in their study exhibit CIL, they need to show that this CIL response is dependent on Semaphorin/CIL. Loss of function experiment of these molecules would be the most straight forward approach. Without this basic characterization of CIL the author cannot describe the response trigger by Plexin activation as CIL.

As suggested by the reviewer, we conducted additional experiments to characterize the CIL behaviour of primary osteoblasts and osteoblastic MC3T3-E1 cells. These additional experiments include co-culture and wound healing assays of primary osteoblasts and primary osteoclasts (supplementary Fig. S1a,b, supplementary movie S1, S2), and co-culture/wound healing assays of osteoblastic MC3T3-E1 cells or its Plexin-null derivatives (see below) with primary osteoclasts (Fig. 1a-g, supplementary Fig. S4c,d, supplementary movies S3, S5-S7). Our observations, which were consistent with CIL, were characterized in Fig. 1f,g and supplementary Fig. S2a,b. These new results provided much stronger evidence on CIL between osteoclasts and osteoblasts.

In loss-of-function studies, we initially set off to perform knockdown using either pooled siRNA oligos or shRNA-expressing lentiviruses. The reagents and knockdown results were described in Page 6, Methods and supplementary Fig. S3d,e. Both siRNA and shRNA approaches worked, but the knocked down efficiencies with neither approach were stellar, with at best at a 70% knockdown in western blot. The toxicity associated with transient transfection (siRNA) and high viral titers (lentiviral shRNA) also discouraged us from conducting migration assays on these cells. We alternately took a CRISPR/Cas9 approach to perturb the endogenous Plexin-B1. We transduced MC3T3-E1 cells with Cas9 and two targeting guide RNAs that are flanking the start codon of the PLXNB1 gene (Supplementary Table T1, supplementary Fig. S3f). This allowed us to precisely delete only a short stretch (~100 bp) of PLXNB1 gene (supplementary Fig. S3g) while completely eliminating Plexin-B1 translation (Fig.1d). Two independent knockout lines were generated and verified by DNA sequencing. In migration assays, both KO lines showed a clear reduction of CIL when their collapse response upon contact with osteoclasts and the distance by which the cells were separated after contact were measured (Fig. 1e-g, supplementary movies S6, S7). In addition, we conducted co-culture/wound healing assays using MC3T3-E1 cells with undifferentiated BMMs that lack Sema4D and found a defect in CIL as well (supplementary Fig. S4a,b, movie S4). Thanks to the reviewer's suggestion, we were able to attribute the CIL in osteoblasts to Plexin-B1 signaling with these new data, which provides a better physiological context for our development of optoPlexin.

2. The characterization of CIL presented in this study is not adequate. CIL involves cell

migration and not only collapse of cell protrusions. The authors should perform longer movies to see whether localized activation of Plexin leads to a change in the direction of cell migration, as expected if CIL is involved. Velocities before and after cell collision and acceleration should be determined. In addition, these changes in cell migration should be compared with the changes induced by proper cell-cell collision (point 1). The data (and movies) presented are too short to see a proper change in the direction of migration, as it is only possible to appreciate a change in the orientation of protrusions.

We analyzed these additional migration experiments described above to obtain velocities of leading edge, centroids and nuclei of the cells, and to extract timing data on retraction and distal protrusion. These new data were presented in Fig.4d,e, and supplementary Fig.S2a. Specifically to the reviewer's request, we performed longer time lapse (30 min-1 hour, imaging was stopped when the cell migrated out of the field of view) imaging with optoPlexin to be certain of its impact on cell migration. These data are presented in Fig.S16 and movie S12. In the movie, in addition to the reorientation of protrusions, one can clearly see the change of migration direction upon optoPlexin activation and the displacement of the cell. Upon analyzing the velocity data, we failed to detect any significant changes in centroid velocity or nucleus velocity (Fig.5f, supplementary Fig. S2b, S17). However we clearly observed the rapid retraction of the leading edge in both wild type cells and optoPlexin cells.

3. The discussion is rather poor and there are aspects of CIL that do not completely fit with the data presented here. For example, it has been recently shown that CIL involves

generation of tension across the cell-cell contact, and this could be an important aspect of the CIL response (Davis et al., 2015; Cell 161, 361). However, the "CIL response" induced in this manuscript cannot involve tension across the cell-cell contact. In addition, it has been shown that during CIL the protrusions away from the cell contact are formed before cell-cell separation (Scarpa et al., 2015; Dev Cell 34, 421), but in this manuscript the authors show that the protrusions away of the cell contact are produced after cell contraction in the region that would correspond to the cell-contact (activation of Plexin). The authors need at least discuss their findings in relation to what is known about CIL.

We agree that our discussion in the original manuscript was rather poor. We have since significantly revised this section and expanded discussion on CIL. Specifically, the relevance of cell-cell contact in CIL in osteoblast and the limitations of optoPlexin were discussed. Regarding the sequence of retractions and protrusions, we wanted to clarify that we observed the initiation of retraction to precede formation of protrusions. We defined initiation of retraction as the time point when the protrusions in contact with osteoclasts or where optoPlexin was stimulated just began to retract. There is a significant delay between this initiation and complete retraction (or separation), during which the distal protrusions are initiated and stabilized to significantly alter the direction of migration. In our co-culture experiments, distal protrusions are formed before osteoblasts became separated from osteoclasts (see Movie S2, S3 and S5). Our observations are indeed consistent with previous reports in that it is these newly form protrusions that enable the cell to migrate away from the site of contact (or optoPlexin activation) and thus mediate cell-cell separation.

4. In Fig 2f, the levels of Plexin activation do not coincide with the levels of RhoA activity: protrusions at the right-bottom corner exhibit low levels of Plexin and high levels of RhoA activity. This is not consistent with the hypothesis that activation of Plexin leads to an immediate activation of RhoA.

In Fig. 3f (originally Fig. 2f), the reviewer correctly pointed out that the intensity of optoPlexin and levels of RhoA activation are poorly correlated. We want to firstly explain that the experiments were conducted with global, rather than local, activation of optoPlexin. As shown in Fig. 2c, global illumination induced less robust translocation to the plasma membrane than local illumination did. Obviously the data would have been more convincing if we had generated local optoPlexin activation and local RhoA activation. Because of a technical issue--wavelength overlapping with the RhoA sensor, it is difficult to achieve local activation of optoPlexin while imaging the RhoA sensor for the entire cell. In our original submission, we also tried to clarify this matter in supplementary Fig. S14a (new figure label). Both optoPlexin and RhoA activation are clearly induced but their relative changes do not show any clear spatial pattern. The slightly elevated activation of RhoA around the cell periphery may be due to additional factors such as RhoA membrane localization and presence of Rnd1/Rac1 which regulates RhoA activation by Plexin-B1 (Oinuma et al., J Biol Chem. 2003; PMID: 12730235). In our later experiments of local activation of optoPlexin, myoRLC accumulated (Fig.6c-e, supplementary movie S16) around the site of illumination, supporting that RhoA is activated downstream of optoPlexin.

Reviewer #2

Optogenetics has been increasingly emerging as a powerful approach for basic investigation of cellular signaling, particularly in situations where spatial and temporal variation of signaling plays an important role in cell behavior. This manuscript utilizes Optogenetics to study cell protrusion and migration in response to semaphorin-plexin signaling. Specifically, investigators leverage prior findings in the field that Cry2 can both heterodimerize (with CIB1) and homodimerize in response to blue light, by fusing the endodomain of plexins to Cry2 and tethering CIB1 to the cellular membrane. Illumination induces plexin membrane localization and oligomerization, leading to a range of downstream responses including RhoA activation, cell process retraction, process protrusion at a distal site, and redistribution of Rac1/Cdc42/beta-Pix from the retracting to the protruding sites. The placement of a plexin under light control is novel, the experiments are well-designed, the data are high in quality, and the suggestions of downstream mechanism are intriguing. There are only several questions.

We appreciate the reviewer's generous comments on our approach. Some of the points the reviewer brought up are what we were trying to accomplish and couldn't be better said.

If the intracellular domain of a signaling receptor is oligomerized, it is not entirely clearly that membrane localization would be required for activation. For example, in other systems involving optogenetic activation of receptors (e.g. LRP6) activation can occur

without membrane localization. Can the authors speculate why membrane localization would be required for signal activation in this system? Are the effectors investigated in this study (e.g. GEFs) also membrane-localized?

The reviewer brought up an important point. The second design strategy i.e. plasma membrane translocation may be unique for signaling targets that are sensitive to membrane localization. The better known downstream target of Plexin-B1, RhoA for example, is regulated by membrane localization. RhoA is shielded by GDI in the cytosol and can only gain access to GEFs on the plasma membrane. In the literature studies have been using plasma membrane translocation of the catalytic domain of GEF to regulate RhoA activity (van Unen et al., Sci Rep. 2015; PMID: 26435194). More specifically, plasma membrane translocation of both PRG and LARG has been shown to regulate their activation (Carter et al., J Biol Chem, 2014; PMID: 24855647). In addition, we found that mere oligomerization (by omitting CIB-CAAX) is not sufficient for inducing binding of PRG, a GEF for RhoA (Fig. 3d,e). Although further studies are needed to understand the exact mechanism, it is likely that additional factors on the plasma membrane may participate in recruiting PRG, for example Rnd1, which binds to RBD domain of Plexin-B1, has been shown to regulate Plexin signaling. In the revised manuscript, we have provided a more clear rationale for membrane localization and discussed the interesting observation on the requirement of membrane localization (page 18).

For optoPlexin expressing cells, the average time between signal activation and

process retraction was 2.5 minutes. How does (and other aspects of downstream signal activation) quantitatively compared to the local administration of the Sema ligand?

Clearly optogenetic activation affords broader control of pathway activation as a function of space and time, but it would help to benchmark relative to the natural signal a bit more.

Thanks to the excellent suggestion by the reviewer, we made further efforts to extract timing data associated with CIL in cell-contact, beads and optoPlexin-initiated signaling. These results were summarized in Fig. 9h. We also added new data to draw a contrast on the limitations of spatial control and activation of Plexin-B1 in approaches of activating Plexin-B1 using soluble Sema4D ligand and immobilized ligand on silica beads (supplementary Fig.S5-S9). The beads experiments provided data on the kinetics of Plexin-B1 clustering, which is important for downstream signaling, and might explain why osteoclasts and osteoblasts exhibit sustained contact prior to separation. We hope these new data may help to highlight the benefit of using the optogenetic approach developed to complement ligand and cell-cell contact based approaches.

While Figure 3 shows coincident retraction and protrusion in a number of cells, it is unclear what fraction of cells with focally illuminated and retracting processes had a subsequent protrusion. Was it 100%?

The “coincident retraction and protrusion” is highly reproducible in optoPlexin stimulated cells. In the graph for the old Fig. 3 (new Fig. 4d, e, 5d-h), we have provided the number

of cells used in the timing study in the figure legends. In comparison, we observed similar response from osteoblasts when they were in contact with osteoclasts; osteoblasts retracted and moved away in our experiments. The statistics on the endogenous system is shown in Fig. 1f,g.

The relationship between beta-Pix, RhoA, Rac1, Cdc42, and plexin is unclear. There are a number of studies indicating that RhoA, Rac1, and Cdc42 can antagonize one another, such that RhoA activation alone could be hypothesized to contribute to the deactivation of Rac1 and Cdc42. Rather than examine this possibility, the investigators investigate beta-Pix, which clearly does exhibit an interesting depletion in illuminated regions, which could contribute to the coincident depletion of Rac1/Cdc42 activity and localization in such regions. However, it is unclear what causes beta-Pix to become depleted in response to plexin activation. Can the authors comment further on this, as well as the relationship between RhoA and Rac1/Cdc42 activity?

In sum, the optoPlexin design and subsequent logical sequence of experiments to begin to apply this tool to study downstream signaling is elegant. Addressing several questions, and as a byproduct adding more depth to the Discussion, would benefit the manuscript.

As suggested, we have provided further discussion on how crosstalk among Rho GTPases may facilitate the repolarization and on how the translocation of beta-Pix may be regulated. Beta-Pix is one of the important molecules that mediate the cross-talk

between RhoA and Cdc42/Rac1 (Guilluy et al, Trends Cell Biol. 2011). Myosin II-mediated contractility, downstream of RhoA/ROCK, has been shown (Kuo et al., 2011) to be responsible for dissociation of beta-Pix from adhesions and downregulation of Rac activation. We demonstrated that the same mechanism was used downstream of Plexin-B1 signaling; crosstalk between RhoA and Cdc42/Rac1 indeed was taking place here and this cross-talk was mediated by spatial regulation of beta-Pix. Importantly, the new finding in our study is that beta-Pix was not merely depleted by localized Plexin-B1 activation, but were redistributed to distal regions of the cell for activating Cdc42 and promoting new protrusions. In terms of mechanism, we identified ROCK kinase activity being critical for local depletion of beta-Pix. The exact molecular mechanism by which beta-Pix is regulated demands further investigation and is beyond the scope of current study. We have included these points in the final discussion.

Reviewer #3

This article reports the development of a photo-activation system for the cell surface receptor PlexinB1. Results show that upon blue light illumination, the optoPlexin reagent is recruited at the plasma membrane together with the RhoGEFs PRG and LARG, which interact with the C-terminal PDZ binding motif of PlexinB1. FRET experiments revealed that activity of optoPlexin mediates activation of RhoA, a known downstream effector of PlexinB1. The authors then used this system to study the effect of local activation of PlexinB1 in a murine osteoblastic cell line (MC3T3-E1). They observed a cellular response that is reminiscent of contact inhibition of locomotion (CIL) - with retraction of the illuminated protrusion and extension of a new protrusion away from the site of illumination. This response is dependent on the RhoA/ROCK signaling, known to

negatively regulate protrusion formation, but also involves the GAP activity of PlexinB1. Concomitant to retraction, the authors observed a repolarization of PIP3, cdc42, Rac1 and the RhoGEF beta-pix at the new protruding front.

Overall the data are interesting and convincing, although I feel that the behavior of cells that follows photo-activation of optoPlexin could have been more extensively documented. For example, the statement that « morphological changes altered the migration direction and caused the cell to migrate away from the site of illumination » (line 223-224) is not demonstrated (Fig.4 only shows that cell velocity remains unchanged).

We appreciated the reviewer's positive comments on our study. As suggested, we have provided more characterization of optoPlexin cells as illustrated in new figures (Fig. 5c, supplementary Fig.S16a-c, S17, supplementary movie S12). In particular, the concern on how the optoPlexin cells migrate after illumination was better illustrated in supplementary Fig.S16a-c and movie S12.

One may wonder, however, how relevant these findings are for the biology of osteoblasts. To what extent does photo-activation of optoPlexin mimic an interaction with Sema4D-expressing osteoclasts? The authors should provide evidence that heterotypic CIL can occur between osteoblasts and osteoclasts and that it is mediated by Sema4D/PlexinB1 signaling. It is important because a role of the Sema4D/PlexinB1/RhoA/ROCK signaling axis has been previously reported by Negishi-

Koga et al. to stimulate motility of osteoblastic cells. How do the authors reconcile this result with their findings of local retraction/collapse after optoPlexin activation?

The previous report by Negishi-Koga et al. (Nat Med. 2011; PMID: 22019888) attracted huge interest in Plexin signaling and suggested that targeting semaphoring 4D may provide new therapeutics for osteoporosis. For molecular mechanisms, the study concluded that Plexin-B1 signaling stimulates osteoblast motility but provided limited explanation on why osteoblasts are in close contact with osteoclasts in Plexin-B1 deficient mice. It has always been thought that osteoclasts attract osteoblasts. Under such paradigm, disrupting Plexin-B1 and compromising motility would have resulted in greater space between these two cell types. Yet it is in the wild type mice, where Plexin-B1 is functional, the space between osteoclast and osteoblast is maintained. In the revised manuscript, we add new data on how osteoclast repels osteoblast in a manner that is dependent on Plexin-B1. These new data are presented in Fig. 1a-g, supplementary Fig.S1, S2, S4 and supplementary movie S1-S7. Our study provides a fresh interpretation of how osteoclast may affect osteoblast migration. Osteoclasts may attract osteoblast in distance through chemotaxis but repel osteoblasts upon contact. Such a complex communication between these two cell types may be critical for the coupling of their actions for efficient bone remodeling. Additionally, loss of Sema4D has been reported to impact bone resorption (Dacquin et al., PLOS ONE, 2011; PMID: 22046317). It would be interesting to test whether similar signaling mechanisms are present in osteoblastic bone lining cells, which osteoclasts might utilize to gain access to the bone matrix to initiate resorption.

What I worry about is that important co-receptors for PlexinB1 function (such as ErbB2) may not be recruited and/or trans-activated by the optoPlexin construct. Therefore, the biological outcome of optoPlexin signaling would have little physiological relevance, at least in the context of the study of osteoblastic cells.

We agree with the reviewer that optoPlexin most likely does not form complex with ErbB2 and therefore may not affect ErbB2 signaling, since the interaction are reported to be mediated by the extracellular domains of Plexin-B1 and ErbB2. ErbB2 can certainly have additional impact on osteoblasts. However, as shown by Driessens et al. (Curr Biol. 2001; PMID: 11267870), clustering of the cytosolic domain of Plexin-B1 on the membrane is sufficient for RhoA activation. Our results also demonstrated that optoPlexin sufficiently recruits RhoGEF and activates RhoA. ErbB2 are known to regulate Plexin-B1 by phosphorylating two tyrosine residues, Y1692 and Y1716. Their phosphorylation was thought to mediate the recruitment of PLC-gamma, which leads to binding/activation of PRG. To specifically address whether such interaction is essential for optoPlexin-mediated RhoA activation, we generated point mutations in optoPlexin in which the two Tyr residues are mutated to Phe (optoPlexin-YF, supplementary Fig. S10) and saw no defects in recruitment of PRG and RhoA activation (supplementary Fig. S15b, c). Additionally, optoPlexin-YF induced identical morphological changes in MC3T3-E1 cells and optoPlexin (supplementary Fig.S19, movie S18). We also did not observe any effect of erlotinib, an inhibitor ErbB2, on PRG recruitment with optoPlexin (supplementary Fig. S15). Thus we at least can conclude that optoPlexin are sufficient

in activating RhoA pathways and elicit CIL-like response in the absence of ErbB2 activity.

Another major concern that should be addressed is that, despite evidence for a spatial redistribution of several molecules contributing to polarized cell migration (cdc42, Rac1, Pip3, beta-PIX), this study does not provide functional evidence for their implication in PlexinB1-mediated CIL, nor on how PlexinB1 signaling may be mechanistically coupled to this polarity switch.

Beta-Pix has been shown to be a critical regulator of osteoblast migration (Kutys and Yamada, Nat Cell Biol. 2014; PMID: 25150978). To pinpoint a sole functional significance of beta-Pix in CIL can be challenging as perturbing its activity can drastically compromise cell migration. We tried to provide indirect evidence in Fig. 9g by titrating the expression levels of beta-Pix while gauging the CIL response after optoPlexin activation. Our data is consistent with redistribution of beta-Pix being a prerequisite of CIL initiation. In terms regulatory mechanism, we have shown in Fig. 9f that RhoA-ROCK signaling initiated by optoPlexin appeared to be required for the redistribution of beta-Pix.

Minor points:

Control experiments showing membrane recruitment of optoPlexinRA are missing.

We have provided the results of the control experiments on optoPlexin-RA in supplementary Fig.S12, S18 and movie S17.

The article by Polleux et al. reports a role of Sema3A gradient on the patterning of apical dendrites of cortical pyramidal neurons, not on "outward radial migration of neurons in a developing cortex" (line 48-49)). Line 46 writes "Semaphorins from family 3 and 7 are secreted (...)". Actually, Sema7a is not secreted but is a glycosylphosphatidylinositol (GPI)-anchored membrane glycoprotein.

We apologize for having incorrectly referenced Dr. Polleux's work and for the incorrect statement on Sema7a. These issues have been addressed in the revised Introduction.

The figure legends for Figures 8f and 8g are swapped (p.42).

We have corrected the figure legends for Fig. 9f and g (old Fig. 8).

Reviewer #4:

The authors use an optogenetic approach to achieve localised activation of a truncated Plexin B1 construct. They show that localised activation of Plexin B1 causes localised cell retraction. Given that global activation of the Plexin is known to give global retraction, this is somewhat predictable.

We agree that the local retraction phenotype induced by optoPlexin is somewhat

predictable given vast literature have demonstrated a cell collapse phenotype with soluble semaphoring ligands. What was not expected in our study is the protrusion at distal regions induced immediately after initiation of retraction downstream of optoPlexin activation. Local activation of Plexin-B1, as demonstrated using optogenetics, can impact cell migration in a more profound way. Rather than simply collapsing the cell and waiting for other unknown factors to induce migration, activation of Plexin-B1 alone is sufficient to induce repolarization of the cell, which is consistent with the role of Semaphorins in acting as guidance molecules. This occurs with predictable timing in both wild type cells and the optoPlexin cells. We propose that this is due to the ability of Plexin-B1 to redistribute an activator(s) of protrusion. We identified the beta-Pix is one of the activators that are regulated in such. In addition, unlike global activation, localized Plexin-B1 signaling did not hinder the migratory capability of the cells, but altered the direction of migration, which allowed us to probe some of the signaling pathways through which repulsive guidance cues might act.

The authors describe the localised optical activation of Plexin B1 as contact inhibition (CIL). While these assays may mimic CIL, they are not replicating the conditions of CIL. The truncated Plexin has no transmembrane domain and does not see an immobilised semaphorin on an adjacent cell. It is difficult to see what advantage the optogenetic approach gives here, compared with looking at actual CIL with another cell.

We appreciate the reviewers comments, which prompted us to perform additional experiments to get a more complete picture of the signaling, involved in Plexin-B1

mediated CIL between cells. We recognize the flaw by omitting cell contact in studying CIL. In addressing this concern, we have provided a new set of data on Plexin-B1-mediated CIL where osteoblasts were in contact with osteoclasts. These new data are presented in Fig. 1, supplementary Fig.S1, S2, S4 and supplementary movie S1-S7. In these experiments we were able to obtain data on cell morphology and motility using DIC/phase contrast imaging. However, imaging signaling dynamics in osteoblasts during CIL induced by osteoclasts turned out to be technically challenging. BMMs do not differentiate very efficiently to osteoclasts on glass surface which is critical for high resolution microscopy. Additionally, multinucleated osteoclasts are relatively short lived *in vitro* (15-20 hours) and it is also difficult to lift mature osteoclasts and reseed them to imaging chambers. In a culture sparse enough to observe single cell-cell encounters, osteoblast-osteoclast contact is fairly infrequent. Given the transient nature of their contact, it is difficult to employ high-resolution live cell imaging to investigate such encounters. These technical issues prevented us from imaging in true CIL most of the signaling dynamics reported in the manuscript. CIL has a much broader context and can be initiated with diverse upstream signaling. In this case we were particularly interested in Plexin-B1 signaling in CIL. We agree that optogenetic approach alone would not provide any specific information on several aspects of Plexin-B1 signaling in CIL such as receptor clustering, the nature of cell-cell contact or the timeline of separation of the cells. We attempted to probe some of these questions using co-culture assays and immobilized Sema4D-Fc experiments. The challenge is, however, that there is no existing method to efficiently interrogate localized Plexin signaling. We believe that the optogenetic approach complements our cell co-culture and ligand based experiments to

elucidate the intracellular spatial dynamics of different migratory factors in cells undergoing CIL.

The authors extend their imaging based analysis to analyse Rho GTPase signalling downstream of localised Plexin activation. They make some useful observations about localised Rho signalling; however, these are either based on known Plexin signalling or broadly predictable from the wider literature on cytoskeletal signalling.

As one of the earlier reviewers pointed out, the unique benefit of optogenetics is precise spatial and temporal control. In our case, only with optoPlexin, we could specifically and locally activate the receptor. Doing so also demonstrated causality in that Plexin-B1 activation alone is sufficient for inducing CIL. In addition, without using optoPlexin, we would not have identified that Plexin can spatially regulate beta-Pix and sufficiently induce cell repolarization without inducing cell collapse. Using optoPlexin, we also for the first time mapped out the temporal kinetics of signaling cascades downstream of Plexin-B1 during CIL along with the kinematics of cell migration. These are some of the key findings of the manuscript. We hope in the revised version they are more clearly presented to the readers.

Overall, the study is well done and the experimental work is of a good standard. I think that this is publishable work; however, I do not think it meets the criteria for publication at this level. My major criticism is that the optogenetic approach requires compromises without giving benefits. There is no reason not to study actual CIL with a full length

Plexin. Secondly, although the signalling work is valuable, the advance is not enough for publication at this level.

We appreciate the reviewer's generous comments. We thought that some of points we made earlier are reasonable arguments for us to take the optogenetic approach to study Plexin-B1 signaling in CIL. Using OptoPlexin allowed us to overcome some of these technical challenges and enable us to investigate Plexin-B1 signaling during CIL to an extent that is otherwise not easily accessible. In our study, not only we developed a new and robust optogenetic reagent but also we defined the spatial and temporal dynamics of Plexin-B1 signaling in CIL. One of the significant finding from the current study is the previously unknown spatial regulation of beta-PIX by Plexin signaling. It provides a plausible explanation of how cells repolarize during CIL which was previously unknown. Additionally it also provides more information on how cross-talk between RhoA and Cdc42 (or Rac1) is mediated in a cell to alter direction of migration. Furthermore we, for the first time, demonstrated that osteoclasts can repel osteoblasts upon contact which is contrary to the general view that osteoclasts attract osteoblasts. This may help us better understand the role of Sema4D-Plexin-B1 signaling in pathological states, and may further fuel the pursuit of treating osteoporosis using Sema4D blocking antibody. The benefits of such treatment may be more complex than simply affecting osteoblast function. Blocking the ability for osteoclasts to repel osteoblasts may also beneficially compromise its bone degrading activity. In summary we hope the reviewer may appreciate some of the new findings and their implications in the current manuscript.

Reviewers' Comments:

Reviewer #1 (Remarks to the Author)

The authors have adequately addressed my previous concerns

Reviewer #2 (Remarks to the Author)

The authors have conducted additional experimental analysis to compare the timeline of optoPlexin vs. sema beads on pathway activation, as well as added depth to the Discussion on mechanisms of downstream Rho/Rac/Cdc signaling. They have addressed the questions of this reviewer.

Reviewer #3 (Remarks to the Author)

The authors have answered most of my earlier comments in a satisfactory manner. However, they did not convincingly demonstrate that CIL can occur between osteoblasts and osteoclasts. In most examples, migrating osteoblasts are engaged in multiple contacts with surrounding cells (eg. Fig 1c,e, Fig S16). CIL should be studied when an isolated osteoblast contacts an osteoclast. Measuring the collapse of protrusions or the distance between the nucleus of the osteoblast and the point of contact with osteoclast are not appropriate methods to quantify CIL. In the example shown in Fig 1c, this distance decreases 30 min after contact while the histogram in Fig1g shows the opposite effect. CIL is better measured by comparing the contact acceleration indices for free moving and colliding cells. In conclusion I am not convinced on the basis of the evidence presented that Sema4D/PlexinB1 signaling regulates CIL.

Reviewer #4 (Remarks to the Author)

Reviewers' comments:

Reviewer #1 (Remarks to the Author):

The authors have adequately addressed my previous concerns

Reviewer #2 (Remarks to the Author):

The authors have conducted additional experimental analysis to compare the timeline of optoPlexin vs. sema beads on pathway activation, as well as added depth to the Discussion on mechanisms of downstream Rho/Rac/Cdc signaling. They have addressed the questions of this reviewer.

Thanks again to the first two reviewers for your helpful suggestions.

Reviewer #3 (Remarks to the Author):

The authors have answered most of my earlier comments in a satisfactory manner. However, they did not convincingly demonstrate that CIL can occur between osteoblasts and osteoclasts. In most examples, migrating osteoblasts are engaged in multiple contacts with surrounding cells (eg. Fig 1c,e, Fig S16). CIL should be studied when an isolated osteoblast contacts an osteoclast.

We appreciate the comments and suggestions previously provided to us by the reviewer.

In addressing those concerns, we conducted additional experiments which made the

manuscript a much improved one. We are sorry to learn that the reviewer remained unconvinced, specifically having a concern on our experimental design and not using isolated cells to study CIL. We agree that using an isolated osteoblast and an osteoclast could potentially make the data easier to interpret. Unfortunately, the isolated osteoblasts and MC3T3 cells in our observation migrate in random directions in the absence of a chemoattractant, making it very inefficient for them to come in contact with the osteoclasts, especially in a sparse culture. MC3T3 cells also make frequent turns in random migration and, as a result, make it difficult to distinguish between a CIL and a spontaneous turn. On the osteoclast side, efficient differentiation of osteoclasts requires high cell density of the precursors. In our hands, we could not successfully lift and reseed multi-nucleated osteoclasts. We hope the reviewer and the editor will appreciate these technical challenges and understand our choice of the “wound healing” setup which can enforce a directional migration of the osteoblasts/MC3T3 cells and increase the chance of contact.

Measuring the collapse of protrusions or the distance between the nucleus of the osteoblast and the point of contact with osteoclast are not appropriate methods to quantify CIL. CIL is better measured by comparing the contact acceleration indices for free moving and colliding cells. In conclusion I am not convinced on the basis of the evidence presented that Sema4D/PlexinB1 signaling regulates CIL.

We agree that the methods we used--measuring the collapse of protrusion and separation of between osteoblasts and osteoclasts—have their limitations. However

similar approaches have been used to quantify CIL (Scarpa et al., *Biology Open*, 2013, PMID: 24143276; Theveneau et al., *Nature Cell Biology*, 2013, PMID: 23770678; Scarpa et al., *Developmental Cell*, 2015, PMID: 26235046). We also agree that measurement of contact acceleration indices (Cx) is a good way of quantifying CIL. We have since conducted Cx analyses on WT (with or without contact), Cas9 control, and the two KO clones. The new data are presented in Fig 1h and i, and the main text (Page# 5 line# 95 and Page# 6 line# 123) and the Methods section (Page# 25 line# 524) were updated accordingly. The results clearly show a reversal of migration direction in MC3T3-E1 cells upon contact with osteoclasts, which was abolished in cells are Plexin-B1 deficient. As a control, these cells have a Cx values approaching zero prior to contact with the osteoclasts. The analyses of using Cx values further validated the presence of CIL between osteoclasts and osteoblasts, and the role of Plexin-B1 in mediating this behaviour. We thank the reviewer for suggesting this additional approach.

In the example shown in Fig 1c, this distance decreases 30 min after contact while the histogram in Fig1g shows the opposite effect.

The reviewer correctly pointed out an apparent discrepancy between the original Fig. 1c and g. We want to explain that the analyses in 1g were conducted at a different time point (40 min after contact) than the time point used in 1c (30 min). The CIL behavior is better illustrated in movies S2 and S3 which are included in both our original and current

submissions. We have reframed these still images in Fig. 1d and e to cover more time points so the behaviour can be better illustrated in the main figure.

Reviewers' Comments:

Reviewer #3:

Remarks to the Author:

The authors have adequately addressed my last concerns.